# Evidence for virus-mediated oncogenesis in bladder cancers arising in solid organ transplant recipients

**Gabriel J Starrett[1]\*, Kelly Yu[2], Yelena Golubeva[3], Petra Lenz[3], Mary L Piaskowski[1], David Petersen[1], Michael Dean[2], Ajay Israni[4], Brenda Y Hernandez[5], Thomas C Tucker[6], Iona Cheng[7], Lou Gonsalves[8], Cyllene R Morris[9], Shehnaz K Hussain[10], Charles F Lynch[11], Reuben S Harris[12], Ludmila Prokunina-Olsson[2], Paul S Meltzer[1], Christopher B Buck[1†], Eric A Engels[2†]**

[1]CCR, NCI, NIH, Bethesda, United States; [2]DCEG, NCI, NIH, Rockville, United States; [3]Leidos Biomedical Research Inc, Frederick, United States; [4]Department of Medicine, Nephrology Division, Hennepin Healthcare System, University of Minnesota, Minneapolis, United States; [5]Cancer Center, University of Hawaii, Honolulu, United States; [6]The Kentucky Cancer Registry, University of Kentucky, Lexington, United States; [7]Department of Epidemiology and Biostatistics,and Helen Diller Family Comprehensive Cancer Center, University of California, San Francisco, Fremont, United States; [8]Connecticut Tumor Registry, Connecticut Department of Public Health, Hartford, United States; [9]California Cancer Reporting and Epidemiologic Surveillance Program, University of California, Davis, Davis, United States; [10]Cedars-Sinai Cancer and Department of Medicine, Cedars-Sinai Medical Center, Los Angeles, United States; [11]The Iowa Cancer Registry, University of Iowa, Iowa City, United States; [12]Howard Hughes Medical Institute, University of Minnesota, Minneapolis, United States

**\*For correspondence:**
gabe.starrett@nih.gov

†These authors contributed equally to this work

**Abstract** A small percentage of bladder cancers in the general population have been found to harbor DNA viruses. In contrast, up to 25% of tumors of solid organ transplant recipients, who are at an increased risk of developing bladder cancer and have an overall poorer outcomes, harbor BK polyomavirus (BKPyV). To better understand the biology of the tumors and the mechanisms of carcinogenesis from potential oncoviruses, we performed whole genome and transcriptome sequencing on bladder cancer specimens from 43 transplant patients. Nearly half of the tumors from this patient population contained viral sequences. The most common were from BKPyV (N=9, 21%), JC polyomavirus (N=7, 16%), carcinogenic human papillomaviruses (N=3, 7%), and torque teno viruses (N=5, 12%). Immunohistochemistry revealed variable Large T antigen expression in BKPyV-positive tumors ranging from 100% positive staining of tumor tissue to less than 1%. In most cases of BKPyV-positive tumors, the viral genome appeared to be clonally integrated into the host chromosome consistent with microhomology-mediated end joining and coincided with focal amplifications of the tumor genome similar to other virus-mediated cancers. Significant changes in host gene expression consistent with the functions of BKPyV Large T antigen were also observed in these tumors. Lastly, we identified four mutation signatures in our cases, with those attributable to APOBEC3 and SBS5 being the most abundant. Mutation signatures associated with an antiviral drug, ganciclovir, and aristolochic acid, a nephrotoxic compound found in some herbal medicines, were also observed. The results suggest multiple pathways to carcinogenesis in solid organ transplant recipients with a large fraction being virus-associated.

## Editor's evaluation

This work provides a compelling case that viral origins of bladder cancer should be more carefully considered. Specifically, the clonogenic expression of viral oncogenes in these tumors combined with the lower than expected prevalence of major tumor suppressor (p53 and pRB) provides strong evidence for the authors' assertions. Ultimately, it will be important to follow up this work and I look forward to seeing those next steps.

## Introduction

At least 20% of all cancers are attributable to viral, bacterial, or parasitic infections (*de Martel et al., 2020*). The advent of high-throughput deep sequencing has provided unprecedented opportunities to learn how infectious agents are involved in cancer in an unbiased manner. Several previous studies have searched for microbial nucleotide sequences in The Cancer Genome Atlas (TCGA), the International Cancer Genome Consortium (ICGC), and PanCancer Analysis of Whole Genomes (PCAWG) datasets (*Zapatka et al., 2020*; *Cantalupo et al., 2018*). In addition to confirming known associations, such as the presence of human papillomaviruses (HPVs) in cervical cancer, these studies also uncovered rare cases in which viral sequences were unexpectedly found in other major cancers affecting the general population (*Cantalupo et al., 2018*).

Despite the immense amount of tumor sequencing data generated to date, the identification of microorganisms in common cancers through these studies has been limited. A more focused assessment of groups at increased risk for virus-associated cancers may be needed. In particular, oncogenic viruses may contribute to a larger fraction of cancer cases among immunosuppressed individuals, such as those with human immunodeficiency virus (HIV) infection and organ transplant recipients. These populations have been previously shown to be at increased risk for developing papillomavirus-mediated cancers, and oncogenic viruses, such as Kaposi's sarcoma-associated herpesvirus and Merkel cell polyomavirus (MCPyV), were discovered in these patient populations (*Chang et al., 1994*; *Feng et al., 2008*; *D'Arcy et al., 2021*).

Roughly a dozen 'high-risk' HPV types cause nearly all cervical cancers, a large majority of other anogenital cancers, and about half of all oropharyngeal cancers (*Graham, 2017*). The carcinogenic effects of these small circular double-stranded DNA viruses are primarily dependent on the expression of the E6 and E7 oncogenes which, among a wide range of other functions, inactivate the tumor suppressor proteins p53 and Rb, respectively (*Rosty et al., 2005*; *Crook et al., 1991*; *Mirabello et al., 2017*; *DeCaprio, 2014*; *Barbosa et al., 1990*).

Polyomaviruses share many biological features with papillomaviruses. In particular, polyomavirus T antigens perform many of the same functions as papillomavirus oncoproteins and are similarly oncogenic in cellular and animal models (*Moens and Macdonald, 2019*). MCPyV has been identified as an etiological factor in a rare skin cancer, Merkel cell carcinoma (*Feng et al., 2008*; *Shuda et al., 2008*; *Kassem et al., 2008*). Another human polyomavirus, BKPyV, has a long-debated history as a candidate cancer-causing virus. Several case reports have described the detection of BKPyV in bladder cancers arising in transplant recipients, and kidney recipients who develop BKPyV viremia or BKPyV-induced nephropathy (BKVN) after transplant are at increased risk of bladder cancer (*Gupta et al., 2018*; *Vajdic and van Leeuwen, 2009*; *Li et al., 2021*; *Papadimitriou et al., 2016*).

Similar to HPV-induced cervical and oropharyngeal cancers, bladder cancers exhibit somatic point mutations that are largely attributable to the activity of APOBEC3 family cytosine deaminases (*Robertson et al., 2017*; *Burns et al., 2013*; *Roberts et al., 2013*). These enzymes normally function as innate immune defenses against viruses by deaminating cytosines in single-stranded DNA, leading to hypermutation of the viral genome (*Poulain et al., 2020*). Commonly, APOBEC3 enzymes, particularly APOBEC3A and APOBEC3B, can become dysregulated and cause carcinogenic damage to cellular DNA during the development of various types of cancer (*Swanton et al., 2015*). APOBEC3A and APOBEC3B are upregulated in response to the expression of HPV E6 and E7, and APOBEC3A can restrict HPV replication (*Mori et al., 2017*; *Warren et al., 2017*; *Ahasan et al., 2015*; *Warren et al., 2015b*; *Vieira et al., 2014*). The large T antigens (LTags) of BKPyV and JC polyomavirus (JCPyV, a close relative of BKPyV) also upregulate APOBEC3B expression and activity (*Starrett et al., 2019*; *Peretti et al., 2018*; *Verhalen et al., 2016*).

**Table 1.** Characteristics of post-transplant bladder cancer cases (N=43).

| Characteristic | Statistic | |
| --- | --- | --- |
| | Median | IQR |
| Age in years at diagnosis | 65 | 60, 71 |
| Years from transplant to diagnosis | 5.8 | 3, 7 |
| | N | % |
| Sex | | |
| Female | 13 | 30 |
| Male | 30 | 70 |
| Transplanted organ | | |
| Kidney | 24 | 56 |
| Liver | 4 | 9 |
| Heart and/or lung | 14 | 33 |
| Pancreas | 1 | 2 |
| Race | | |
| Non-Hispanic White | 30 | 70 |
| Asian/Pacific Islander | 8 | 19 |
| Hispanic | 5 | 12 |
| Summary stage | | |
| In situ | 12 | 28 |
| Localized | 19 | 46 |
| Regional | 7 | 14 |
| Distant | 5 | 12 |
| Grade | | |
| Low | 20 | 47 |
| High | 22 | 51 |
| Papillary urothelial neoplasm of low malignant potential | 1 | 2 |

IQR: interquartile range.

To characterize the mutational, transcriptomic, and viral landscapes of bladder cancers arising in immunosuppressed individuals, we evaluated archived tissues from 43 solid organ transplant recipients who developed this malignancy. We performed total RNA sequencing and whole genome sequencing (WGS) from these tissues. We utilized high-sensitivity methods and comprehensive reference databases of sequences for conserved viral proteins to identify known viral species and to search for divergent viruses. Once viruses were identified, we further evaluated the sequence data for integration events, point mutations, mutation signatures, and differentially expressed genes to identify differences correlating with the presence of these viruses and their integration state.

## Results

### Bladder cancers from transplant recipients

The study population was comprised of 43 U.S. cases from patients who developed bladder cancer after receiving solid organ transplantation (*Table 1* and *Supplementary file 1a*). Seventy percent were male and 70% were non-Hispanic white. The median age at cancer diagnosis was 65 years (range: 27–82). The most commonly transplanted organ was the kidney (56%), followed by the heart and/or lung (33%) and liver (9%). Primary tumors were roughly an equal mixture of high- and low-grade carcinomas diagnosed with a median of 5.7 years after transplantation. Twelve cases were categorized as in situ as defined by the Surveillance, Epidemiology, and End Results (SEER) Program, with two of those cases being transitional cell carcinomas in situ and ten cases being noninvasive papillary transitional cell carcinomas. Invasive cases were mostly categorized into the localized stage (n=20, 46%), which includes tumors that have invaded into the mucosa, submucosa, muscle, or subserosa. The 11 remaining cases either had regional or distant invasion or metastasis. We successfully generated WGS data for 38 primary tumors, three metastases, and 10 normal (histologically non-malignant) tissues, with a median of 31 x coverage across the human genome (range: 14–55 x) (*Supplementary file 1a*). We generated total RNA sequencing data for 43 primary tumors, five metastases, and 14 normal tissues, with a median of 30 million reads per sample (range: 4–65.5 million).

### Detection of viruses in bladder cancers from transplant recipients

Analysis of WGS data for 38 primary tumors identified one or more virus species in 17 specimens (45%) (*Figure 1A* and *Supplementary file 1a*). RNA sequencing on tumor samples for which WGS data could not be obtained revealed three additional cases containing viral sequences (45% of samples overall). Among the 20 virus-positive primary tumors, the majority harbored BKPyV (n=9) or JCPyV (n=7). High-risk HPV genotypes 16 and 51 were detected in one and two tumors, respectively. A

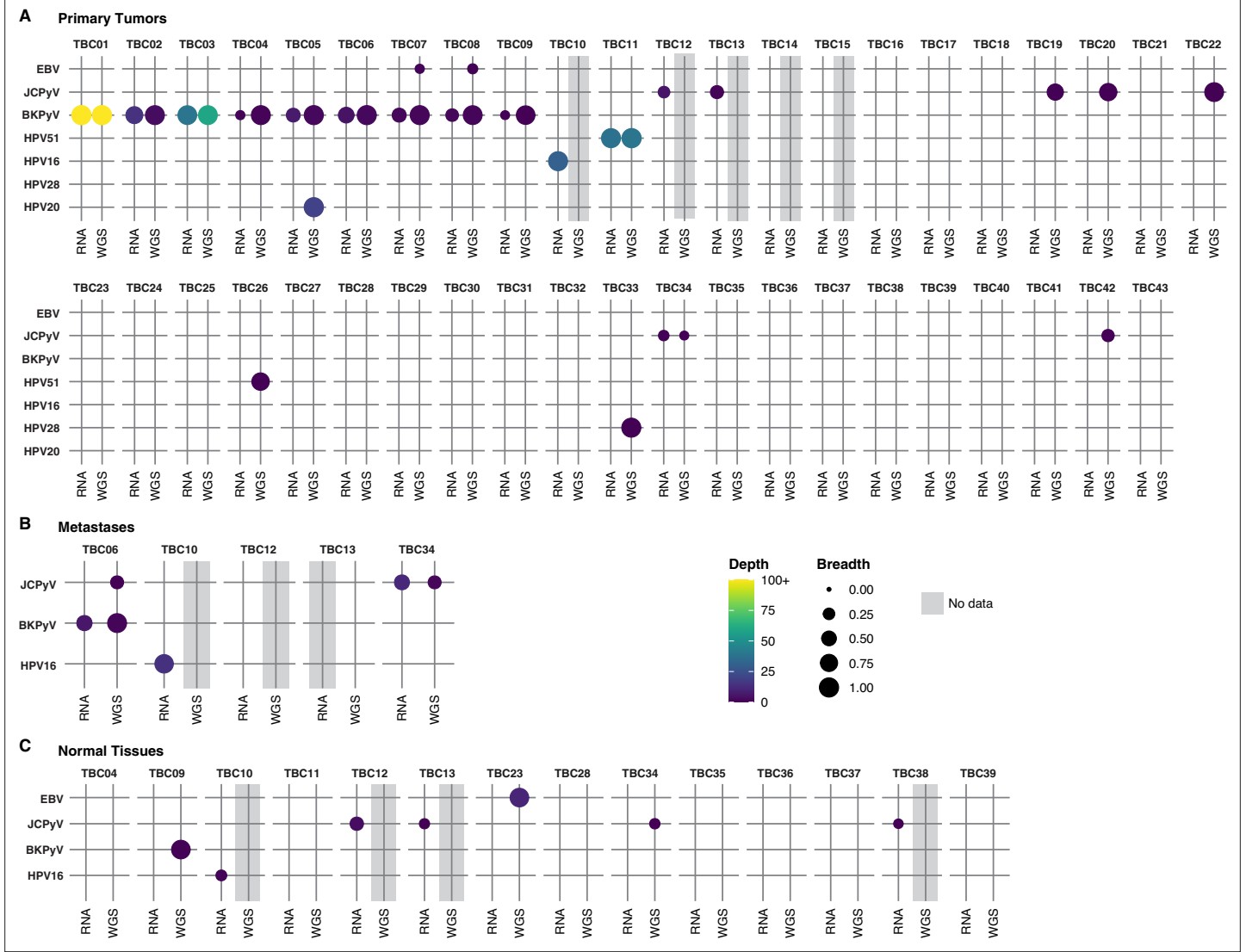

**Figure 1.** Detection of viral sequences. (**A**) Primary tumors. (**B**) Metastatic tumors. (**C**) Normal tissues. Viral species are shown on the rows, and each case in the cohort (represented with a TBC number) is a column. TBC numbers represent a single case and are consistent across primary, metastatic, and normal tissues. Circle size represents the breadth or fraction of the viral genome covered, and color represents the average depth of coverage of the viral k-mers with all coverages over 100 binned together. Specimens without sequencing data have a gray background.

The online version of this article includes the following figure supplement(s) for figure 1:

**Figure supplement 1.** BK polyomavirus (BKPyV) DNA/RNA coverage plots.

**Figure supplement 2.** JC polyomavirus (JCPyV) DNA/RNA coverage plots.

**Figure supplement 3.** All human papillomavirus (HPV) DNA/RNA coverage plots.

low-risk papillomavirus, HPV28, was observed in TBC33. One BKPyV-containing tumor (case TBC05) also harbored relatively abundant amounts of HPV20. Only two reads mapped to HPV20 in the RNA dataset for case TBC05. Sequencing of metastases confirmed the presence of BKPyV in TBC06, JCPyV in TBC34, and HPV16 in TBC10. Additionally, sequencing of two separate tumor sections for TBC03 and TBC09 confirmed the presence of BKPyV in both (*Figure 1—figure supplement 1*).

WGS from TBC16, TBC17, TBC18, TBC19, TBC20, TBC21, TBC22, TBC23, TBC24, TBC27 had low numbers of reads mapping to the BKPyV genome that was judged to be attributable to low levels of index-hopping from TBC01, a papillary urothelial neoplasm of low malignant potential (PUNLMP) that had extremely high BKPyV coverage and was sequenced in the same run. Considering this, along with the absence of RNA reads supporting the presence of BKPyV, we scored these tumors virus-negative.

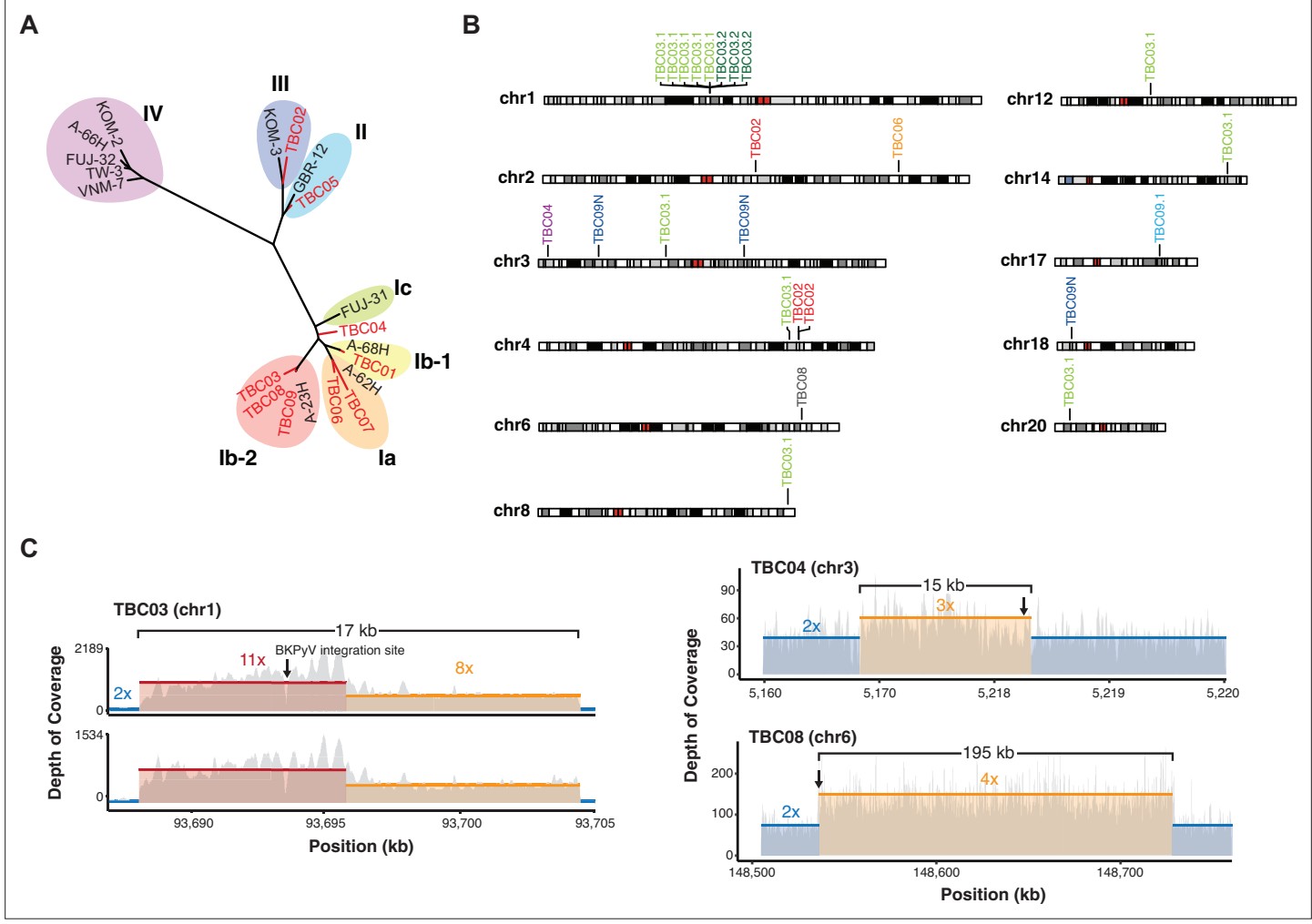

**Figure 2.** Virus diversity and integration. (**A**) Phylogenetic tree of BK polyomavirus (BKPyV) large T antigen (LTag) sequences detected in tumors (red) and reference genotypes with representative strain names. (**B**) Sites of BKPyV integration into host chromosomes are indicated with case numbers. Two separate sections from separate formal-fixed paraffin-embedded (FFPE) blocks of the primary tumor were sequenced for case TBC03 (samples TBC03.1 and TBC03.2). Two separate sections were also sequenced for case TBC09, but an integration site was only detected in sample TBC09.1. Integration sites were also detected in normal tissue sample TBC09N. Black and gray bars indicate cytogenetic bands; red bars indicate centromeres. (**C**) Coverage plot of focal amplifications adjacent to BKPyV integration sites in cases TBC03.1, TBC03.2, TBC04, and TBC08. BKPyV integration junctions are indicated by a black arrow. Colored numbers in the body of the graph indicate coverage depth.

A separate set of searches aimed at identifying divergent members of other virus groups revealed that several tumors (TBC08, TBC14, TBC25, TBC28, and TBC35) and normal tissues (TBC35, TBC28, TBC39) harbored torque teno virus (TTV) sequences from either WGS or RNA sequencing (*Supplementary file 1c*). Epstein-Barr virus was most strongly detected in one normal lymph node (TBC23) and, at low levels, in tumors TBC07 and TBC08. Considering the stronger epidemiological evidence for BKPyV and bladder cancer and its abundance in these specimens, we focused the majority of our analysis on characterizing these tumors.

## Features of BKPyV-positive tumors

BKPyV sequences detected in this study came from every genotype except IV (*Figure 2A*). One patient with a BKPyV-positive tumor had a documented history of BKVN. BKPyV-positive tumors were found in two heart, two lung, one heart and lung, and four kidney transplant recipients. We identified unambiguous BKPyV integration sites in five of the nine BKPyV-positive tumors and in one normal tissue (*Figure 2B* & *Table 2*). For three tumors a single integration junction was identified, and in TBC02 three junctions were identified. In the case TBC03, two separate sections from separate blocks

**Table 2.** BK polyomavirus (BKPyV) integrations sites and microhomology.

| ID | Human sequence match | Virus sequence match | Maximum MH length | MH sequence | Chromosome | Position | Nearest gene (Symbol) | Nearest gene (Ensembl ID) | Distance to Nearest Gene | Nearest RE | Distance to nearest RE |
|---|---|---|---|---|---|---|---|---|---|---|---|
| TBC02 | CATCATGATGATGGG | GATGGGCAGCCTA | 5 | ATGGG | chr2 | 120378301 | INHBB | ENSG00000163083 | −26499 | MIRb | −45 |
| TBC02 | CTCCTGCTCATGAA | CATGAAGGT TAAGCATGCTA | 5 | ATGAA | chr4 | 145732354 | C4orf51 | ENSG00000237136 | 0 | AluSq2 | −474 |
| TBC02 | ACCATTTAATTCCCAA | AGTGGAAATTAC | 2 | AC | chr4 | 145732375 | C4orf51 | ENSG00000237136 | 0 | AluSq2 | −495 |
| TBC03.1 | GCCTTTCTTG TGGACTGGGT | ATTTTCATTTCT ACTGGGGTCAGGA | 0 | No overlap | chr1 | 93693546 | BCAR3 | ENSG00000137936 | 0 | MIRb | 377 |
| TBC03.1 | TCTGTTTCT TATTTCAGAA | GGGTTCTCCTG TTTATAAGGTC | 2 | TC | chr1 | 93693570 | BCAR3 | ENSG00000137936 | 0 | MIRb | 353 |
| TBC03.1 | AGAGCCTTG GTGGTGG | GGTGGCAAA CAGTGCAG | 5 | GGTGG | chr1 | 93693890 | BCAR3 | ENSG00000137936 | 0 | MIRb | 33 |
| TBC03.1 | GATACTTTTT AGACATGC | AACCATGACC TCAGGAAGGA | 4 | CATG | chr1 | 93694075 | BCAR3 | ENSG00000137936 | 0 | MIRb | 0 |
| TBC03.1 | CCTCAAAGC CACCCACTCC | TTTCCATGA GCCCCAAA | 5 | CCAAA | chr1 | 93694843 | BCAR3 | ENSG00000137936 | 0 | MER5A | −92 |
| TBC03.1 | CAATTTTTTTTTTTT | TTTTTTATT TGTAAGGGTG | 7 | TTTTTTT | chr12 | 50449935 | LARP4 | ENSG00000161813 | 0 | AluSc | 0 |
| TBC03.1 | TGCAAGGTG CTTCATGTAT | AGGGGGCTTA AAGGATGCA | 4 | TGCA | chr14 | 95764390 | | ENSG00000257275 | −6735 | MIRb | 0 |
| TBC03.1 | TAGCCAAAA AAAAAAGG | AAAAAAAA GGCCACAG | 11 | AAAAAAAAAGG | chr20 | 8525269 | PLCB1 | ENSG00000182621 | 0 | MamSINE1 | 154 |
| TBC03.1 | CAATTTGGA AAACAAT | ATGCAAGGG CAGTGCACA | 2 | AT | chr3 | 73059264 | PPP4R2 | ENSG00000163605 | 0 | MER103C | 69 |
| TBC03.1 | TAAAAAGTGTCA | AAGTGTCAA TAGAGAAAAA | 8 | AAGTGTCA | chr4 | 142307350 | INPP4B | ENSG00000109452 | 0 | L2a | 0 |
| TBC03.1 | TCACACAAT TT-TACTCCTCT | ACACTTTTTAC ACTCCTCTA | 8 | ACTCCTCT | chr8 | 140923993 | PTK2 | ENSG00000169398 | 0 | L2a | 0 |
| TBC03.2 | GTTGAGTT GGAGCA | CATCTAAATAA TCTCTCAAACT | 2 | CA | chr1 | 93693160 | BCAR3 | ENSG00000137936 | 0 | MER5A1 | −10 |
| TBC03.2 | ACCCAGTCCA CAAGAAAGGC | CCAGTAGAA ATGAAAAT | 0 | No overlap | chr1 | 93693546 | BCAR3 | ENSG00000137936 | 0 | MIRb | 377 |
| TBC03.2 | TCTGTTTCT TATTTCAG | GTTCTCCTGT TTATAAGGTC | 2 | TC | chr1 | 93693570 | BCAR3 | ENSG00000137936 | 0 | MIRb | 353 |
| TBC04 | GAGTGAGT TCATAG | CAACACTGTG GTGAG-TGAGTT | 4 | GAGT | chr3 | 5202593 | EDEM1 | ENSG00000134109 | 0 | L2b | −466 |
| TBC06 | CAGACATT -AGGA | TGAGGACC TAACCTGT | 4 | AGGA | chr2 | 201676427 | MPP4 | ENSG00000082126 | 0 | MIR1_Amn | 0 |
| TBC08 | TCCACTTT CAGTACTT | TGCAAAA AATCAAAT | 1 | T | chr6 | 148535326 | SASH1 | ENSG00000111961 | 0 | AluSq | 995 |
| TBC09.1 | GGGGCGG TAACTAGAAG | ACTAGAAG CTTGTCGT | 8 | ACTAGAAG | chr17 | 61340185 | BCAS3 | ENSG00000141376 | 0 | L2-3_Crp | 0 |
| TBC09N | GAGAAAAT AGGACTCGG | AAGATTCGC CTGAGAAAA | 7 | GAGAAAA | chr18 | 8169205 | PTPRM | ENSG00000173482 | 0 | MER127 | −648 |

*Table 2 continued on next page*

*Table 2 continued*

| ID | Human sequence match | Virus sequence match | Maximum MH length | MH sequence | Chromosome | Position | Nearest gene (Symbol) | Nearest gene (Ensembl ID) | Distance to Nearest Gene | Nearest RE | Distance to nearest RE |
|---|---|---|---|---|---|---|---|---|---|---|---|
| TBC09N | TCCATCC TCCTCTAC | CTCCTCT ACATTGT | 9 | CTCCTCTAC | chr3 | 34028749 | LINC01811 | ENSG00000226320 | 130585 | L2b | 0 |
| TBC09N | ATGTAAT ATAAAACT | CATGATT TTAACCCAG | 0 | No overlap | chr3 | 117678477 | | ENSG00000239268 | 0 | L2c | 0 |

MH: microhomology; RE: Repeat element.

of the primary tumor were sequenced. In 1 of the 2 sections, 11 integration junctions were identified across seven chromosomes. Only three of the junctions could be identified in the second section of the tumor, suggesting either that these junctions were not present throughout the tumor or there was insufficient tumor purity/sequencing depth to detect them.

Integration appeared consistent with a microhomology-mediated end-joining (MMEJ) model for integration, as 20 of 25 junctions (80%) had microhomology greater than or equal to 2 bp. In this model, which has previously been proposed for both HPV- and MCPyV-associated tumors (*Starrett et al., 2020*; *Starrett et al., 2017*; *Akagi et al., 2014*), microhomologies between the virus and host genomes initiate DNA repair processes that can, in some cases, lead to tandem head-to-tail concatemeric repeats of the viral genome as well as focal amplifications of the flanking host chromosome. Consistent with this model, focal amplifications adjacent to BKPyV integration sites were identified in three patient tumors. In TBC03, amplification of a 17 kb region of chromosome 1 flanking a multi-copy BKPyV integrant was observed in two tumor sections (*Figure 2C*). In TBC04, a 15 kb single-copy amplification of chromosome 3 was identified. Lastly, a 195 kb region of chromosome 6 was amplified next to the BKPyV integration junction in TBC08. Twenty-two of the identified 25 junctions (88%) intersected protein-coding genes and thus might conceivably affect gene expression or function.

BKPyV RNA and DNA abundance by sequencing generally did not correspond to specimen tumor purity or the percentage of LTag + cells (*Figure 3A and B*). Gene-level analysis of the RNA sequencing data revealed that 7 of the 9 polyomavirus-positive tumors predominantly expressed the T antigens, with little to no expression of the late genes VP1 and VP2 (encoding the major and minor capsid proteins, respectively) (*Figure 3A and C*, *Figure 3—figure supplement 1*). The LTag open reading frames (ORFs) in these cases were truncated before the helicase domain through deletions, frameshifts, or point mutations. The exception was the BKPyV-positive PUNLMP case TBC01, which showed a balanced expression of both early and late regions.

BKPyV isolates found in cases of polyomavirus nephropathy typically have rearrangements in the regulatory region that enhance viral replication in cell culture. However, in this study, TBC01 was the only polyomavirus-positive tumor showing evidence of regulatory region rearrangements (*Figure 3— figure supplement 2*).

Immunohistochemistry (IHC) for polyomavirus LTag was performed on 18 specimens suspected to contain polyomaviruses and two negative control specimens determined to be free of detectable viral sequences. Control sections were negative for Tag staining, whereas 11 of the 18 specimens that contained polyomavirus sequences showed at least some evidence for Tag positivity (*Figure 3B* and *Figure 3—figure supplement 3*). Three tumors scored as BKPyV sequence-positive had strong to moderate LTag staining in greater than 80% of tumor cells, but the other BKPyV-positive tumors had more variable staining. Moderate to weak staining was visible in less than 0.5% of cells in the primary tumor for TBC06 (*Figure 3D*), but strong staining was observed in about 25% of cells in the metastasis. For TBC09, one sample of the tumor was >90% positive for LTag staining and another sample was less than 25% positive (*Figure 3—figure supplement 3*). The normal tissue for TBC09 showed BKPyV RNA and DNA coverage along a small portion of the regulatory region and small T antigen, but no staining for LTag. Although TBC01 had very high levels of BKPyV DNA and RNA reads, it had the lowest observed proportion of LTag-positive cells (<1% in a section that was >95% tumor as determined by cell morphology). LTag-positive cells in the TBC01 sample were almost entirely localized to the luminal margin of the tumor (*Figure 3D*).

Differential gene expression analysis for BKPyV-positive tumors versus virus-negative tumors revealed 1062 genes that were significantly differentially regulated in tumors harboring BKPyV (*Figure 4A*, *Supplementary file 1e*). Clustering all primary and metastatic tumors by genes with a greater than threefold difference of expression in the above comparison, we identified three major groups that loosely correspond to the amount of BKPyV DNA and RNA in a tumor (*Figure 4C*). A notable exception is the BKPyV-positive tumor TBC01, which falls into the cluster mostly containing virus-negative tumors.

The cluster exclusively containing tumors harboring integrated BKPyV is defined by high expression of genes involved in DNA damage responses, cell cycle progression, angiogenesis, chromatin organization, mitotic spindle assembly, and chromosome condensation/separation as well as some genes associated with neuronal differentiation. Overall, these tumors have relatively low expression of keratins and genes associated with cell adhesion. Genes previously shown to be associated with cell

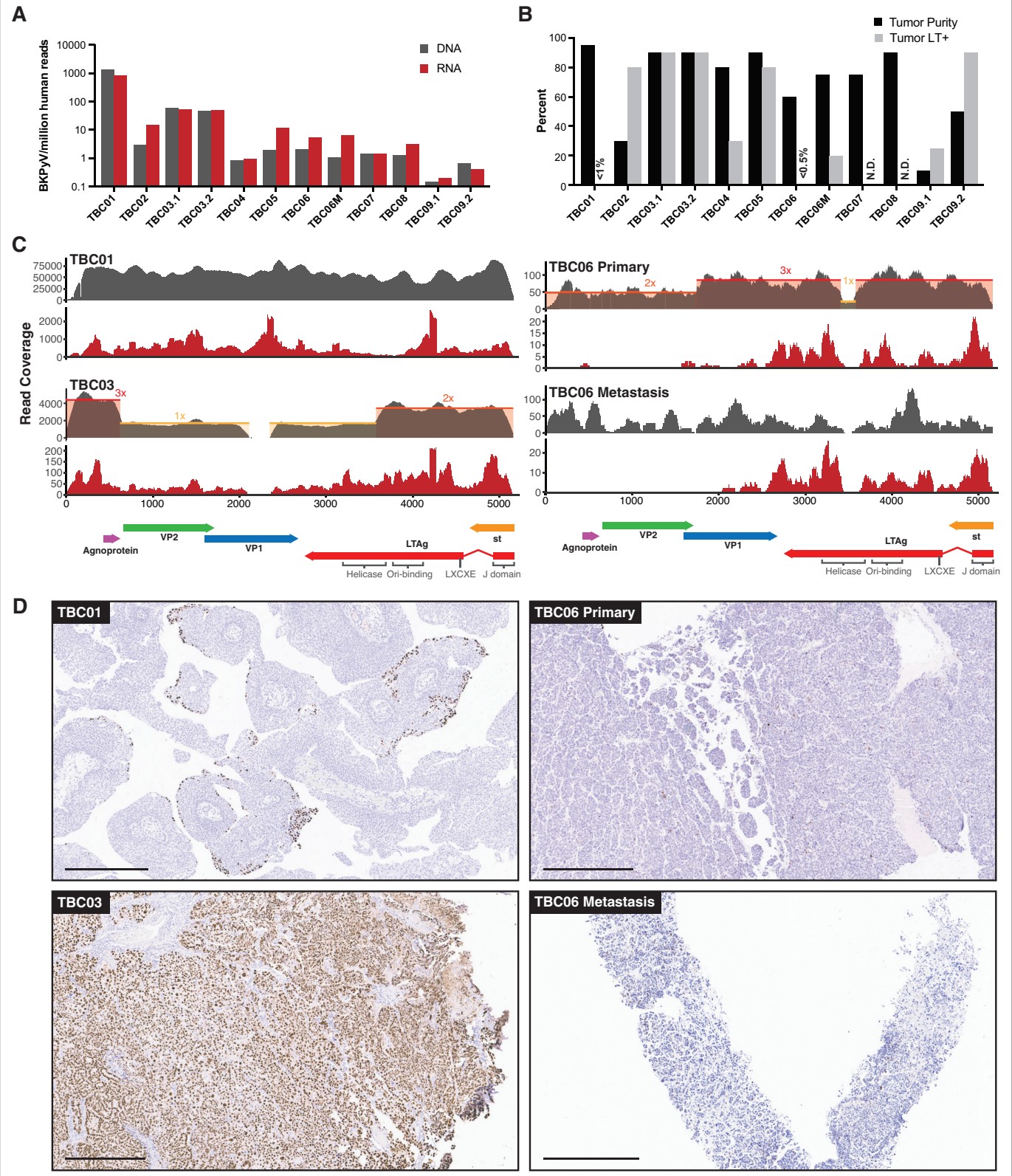

**Figure 3.** BK polyomavirus (BKPyV) DNA, RNA, and large T antigen (LTag) detection in tumors. (**A**) Barplots showing the abundance of BKPyV DNA and RNA reads standardized to human reads (**B**) Barplots of histologically estimated percent tumor purity and Immunohistochemistry (IHC)-positivity for polyomavirus LTag expression. N.D. indicates no IHC image data were generated. (**C**) Representative coverage plots for BKPyV DNA (gray) and RNA

*Figure 3 continued on next page*

*Figure 3 continued*

(red) in BKPyV-positive tumors. Relative copy numbers are indicated by colored boxes and highlight the borders of duplications and deletions in the viral genome. (D) Selected images for LTag IHC highlighting positive staining for BKPyV-positive tumors with scale bars representing 500 microns.

The online version of this article includes the following figure supplement(s) for figure 3:

**Figure supplement 1.** BK polyomavirus (BKPyV) gene expression.

**Figure supplement 2.** Diagrams of the assembled BK polyomavirus (BKPyV) NCCR structures and rearrangements in tumors.

**Figure supplement 3.** T antigen immunohistochemistry (IHC) in BK polyomavirus (BKPyV)-positive tumors.

proliferation in bladder cancer, such as *FGFR3*, had significantly lower expression in BKPyV-positive tumors relative to virus-negative tumors. Notably, tumors harboring BKPyV had significantly higher average *APOBEC3B* expression compared to both normal tissues and tumors not containing any virus (*Figure 4B*). This observation is maintained after stratifying the cases by the germline variant, rs1014971, known to associate with increased APOBEC3B expression and bladder cancer risk with the highest average APOBEC3B expression observed in tumors with both BKPyV and two copies of rs1014971 (*Figure 4—figure supplement 1*).

In TBC03, the observed BKPyV integration into *BCAR3* results in increased expression of the host gene. Further evaluation of RNA reads covering this region revealed a general enrichment of sense and antisense reads mapping to positions 93,688,393–93,704,476, corresponding to the amplified chromosomal region observed in the WGS dataset (*Figure 4—figure supplement 2*). There is an even greater enrichment of mapped reads between positions 93,694,469–93,696,857. No increases in expression in nearby host genes were observed for other cases and integration events.

Aside from integration-related copy number variants (CNVs), large-scale CNVs overall differed between BKPyV-positive and virus-negative tumors (*Figure 5*; *Supplementary file 1g*). BKPyV-positive tumors showed moderate enrichments for gains of chromosome segments 1q, 2 p, 3 p, 7q, 20q, and 22q, while losses of chromosome 2q, 6q, and 10q were also observed more frequently in BKPyV-positive tumors versus virus-negative tumors. Similar differences in copy numbers have been observed for virus-positive and virus-negative Merkel cell carcinoma (*Starrett et al., 2020*).

## Features of other virus-positive tumors

In the cases that were positive for JCPyV, DNA, and RNA coverage depth was much lower than observed for BKPyV-positive tumors, and in several DNA-positive cases, JCPyV transcription was not detected (*Figure 1*). JCPyV reads were detected in three samples from case TBC12 including the primary tumor, tumor-positive lymph node, and adjacent normal bladder wall (*Figure 1—figure supplement 2*). IHC detected sparse LTag staining in JCPyV-positive case TBC13, but not in any tissue samples for TBC12.

For 2 of 3 cases harboring HPV types known to cause cervical cancer (HPV16 and HPV51), transcripts encoding the E6 and E7 oncogenes were detected (*Figure 1—figure supplement 3*). In one HPV16+ case (TBC10), viral oncogene RNA expression was detected in both the primary and metastatic specimens. A possible HPV51 integration event in TBC11 appears to have involved simple repeat sequences and retroelements (*Figure 1—figure supplement 3*). Lastly, one case harbored DNA sequences aligning to HPV20 and a single case harbored DNA aligning to HPV28; however, no RNA reads were detected for these cutaneous HPV types (*Figure 1—figure supplement 3*).

For the five TTV-positive tumors, the WGS analyses did not show evidence of integration. However, we were unable to assemble complete circular genomes for any of the TTVs. The missing segments all overlapped the GC-rich origin of replication that forms stable hairpins and is, therefore, relatively resistant to sequencing with standard Illumina technology (*Tisza et al., 2020*). All observed TTV ORF1 sequences belonged to the *Alphatorquevirus* genus and had 51–100% amino acid identity to previously reported TTV strains (*Supplementary file 1d*).

## Mutation signature analysis

The overall tumor mutation burden, as measured by non-synonymous mutations per million bases, did not show a clear correlation with the presence of viral sequences (*Figure 6A*). We analyzed likely somatic point mutations from all tumors and deconvoluted mutation signatures (*Figure 6B and C*, *Figure 6—figure supplement 1*). As expected for bladder cancer, we commonly observed single-base

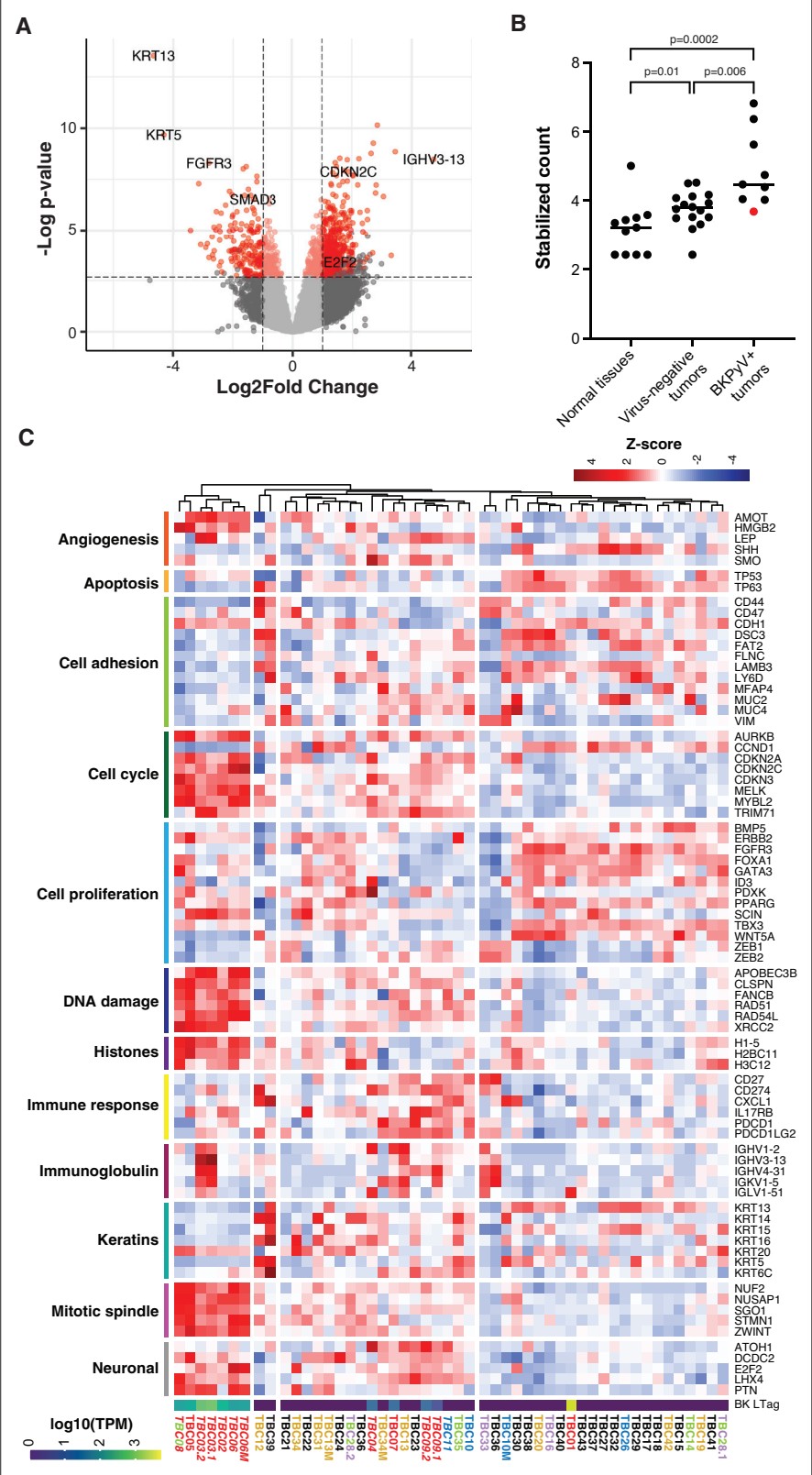

**Figure 4.** Differential gene expression in BK polyomavirus (BKPyV)-positive tumors. (**A**) Volcano plot of differential gene expression between BKPyV-positive and virus-negative tumors. Significantly differentially expressed genes (q-value <0.05, DESeq2) with a fold change greater than two are in red, and genes with a fold change less than two are in pink. Non-significant genes are in gray. (**B**) Variance stabilized counts for APOBEC3B expression from

*Figure 4 continued on next page*

*Figure 4 continued*

DESeq2 grouped by normal tissues, virus-negative tumors, and BKPyV-positive tumors showing significantly increased expression in BKPyV-positive tumors (Mann-Whitney U test). TBC01 is indicated by a red dot. (**C**) Heatmap of Z-scores of significantly differentially expressed genes and genes relevant to bladder cancer grouped by gene ontology. High expression is red, low expression is blue. Tumors names are colored by likely etiology: BKPyV-positive, red; JC polyomavirus (JCPyV)-positive, goldenrod; HR-HPV-positive, blue; torque teno virus (TTV)-positive, green; aristolochic acid, purple; undetermined, black; multiple colors reflect multiple detected viruses or etiologies. Tumors with evidence of integration are in italics. BKPyV LTag expression is shown as log10(transcripts per million [TPM]).

The online version of this article includes the following figure supplement(s) for figure 4:

**Figure supplement 1.** APOBEC3B germline variant and expression by BK polyomavirus (BKPyV) status.

**Figure supplement 2.** Host transcripts from the BK polyomavirus (BKPyV) integration site at BCAR3.

---

substitution 2 (SBS2) and SBS13 (both characteristic of APOBEC3 mutagenesis, N=13 cases) and SBS5 (associated with smoking history and *ERCC2* mutations, N=11 cases).

Four tumors (TBC16, TBC28, TBC31, TBC33) carried a predominant SBS22 signature, which is caused by the chemical aristolochic acid found in the birthwort family of plants. Cases with this signature showed a very high mutational burden (*Figure 6A*). In support of the idea that cases with strong SBS22 signatures arose through environmental exposure, one such case, a kidney recipient, was previously diagnosed with Chinese herbal medicine nephropathy. The final deconvoluted signature closely matched the mutation spectrum caused by the deoxy-guanosine analog ganciclovir, which was recently identified in hematopoietic stem cell transplant recipients (*Figure 6B*; *de Kanter et al., 2021*).

### Recurrent somatic mutations

First, to address the reproducibility of mutation calls in deep sequencing of FFPE samples, we analyzed the sequences from two independent sections from separate blocks for three tumors (*Figure 6D*). Comparing the variants called in these tumors, 77–82% of inferred somatic mutations were common to both sections. Furthermore, a similar comparison showed a large percentage of variants in common between primary tumors and their metastases (*Figure 6D*). In TBC06, 84% of the likely somatic mutations in the metastasis were found in the primary tumor, whereas only 28% of the likely somatic variants in the primary tumor were found in the metastasis. In an additional primary-metastatic pair (TBC34), we identified a similar proportion of shared 'trunk' mutations but the metastasis had more unique, likely somatic variants.

Numerous cellular genes were found to recurrently harbor nonsynonymous, nonsense, and frameshift mutations (*Figure 6E*). The spectrum of frequently mutated genes is similar to those reported in various types of urothelial carcinoma (e.g. mutations in *KMT2D*, *KDM6A*, and *ARID1A*) (*Supplementary file 1f*; *Robertson et al., 2017*; *Nassar et al., 2019*; *Su et al., 2021*). No nonsynonymous mutations were identified in *FGFR3* or *PIK3CA*, even though these genes are commonly mutated in non-muscle invasive bladder cancer (NMIBC) (*Cancer et al., 2014*). Mutations in *TP53*, which are common in muscle-invasive bladder cancer (*Robertson et al., 2017*), were detected in four tumors (*Figure 6E*). None of the HPV-positive tumors with WGS harbored mutations in *TP53* or *RB1*. Similarly, none of the polyomavirus-positive tumors harbored mutations in *RB1* and only TBC08 had a frameshift mutation in *TP53*.

## Discussion

This report presents a comprehensive molecular assessment of 43 bladder cancers arising in solid organ transplant recipients by WGS and total RNA sequencing. DNA and/or RNA sequences of human BK or JC polyomaviruses were detected in 16 tumors (37%). Expression of the polyomavirus LTag was documented immunohistochemically in 10 cases. HPV sequences were detected in six cases, including four cases with HPV types known to cause cervical cancer. Overall, this is a much higher frequency of small DNA tumor virus sequence detection compared to prior surveys of bladder cancers affecting the general population, where fewer than 5% of tumors harbor small DNA tumor virus sequences (*Cantalupo et al., 2018*; *Llewellyn et al., 2018*). The results suggest that human polyomaviruses

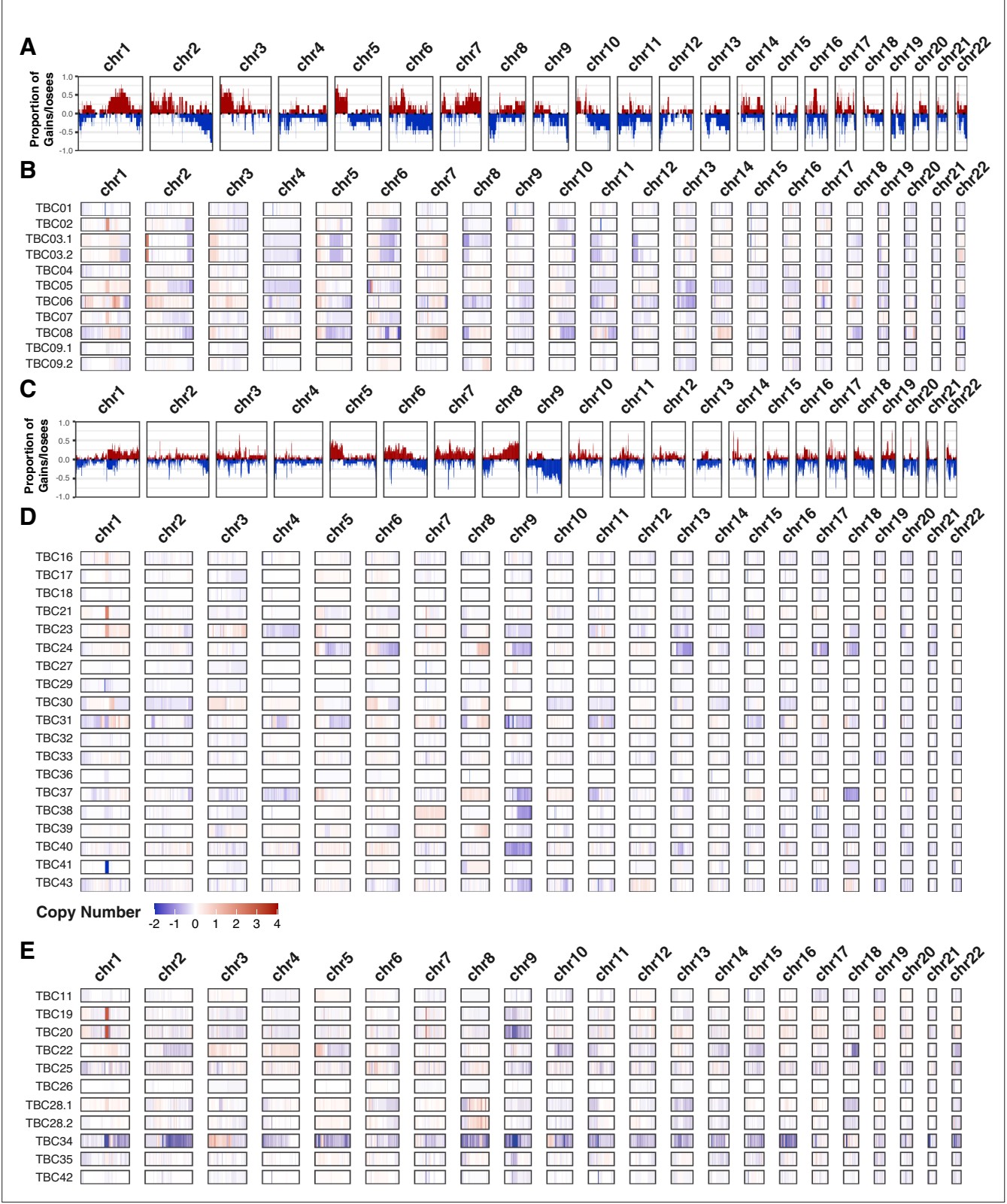

**Figure 5.** Copy number variants. Frequency plots for large copy number variants in BK polyomavirus (BKPyV)-positive tumors (panel **A**) and virus-negative tumors (panel **C**). Frequency of gains/amplifications is shown in red; losses/deletions are shown in blue. Sample level copy number variant spectra for BKPyV-positive tumors (panel **B**), virus-negative tumors (panel **D**), and all other tumors (panel **E**). Complete deletions are in dark blue and high copy amplifications are in red.

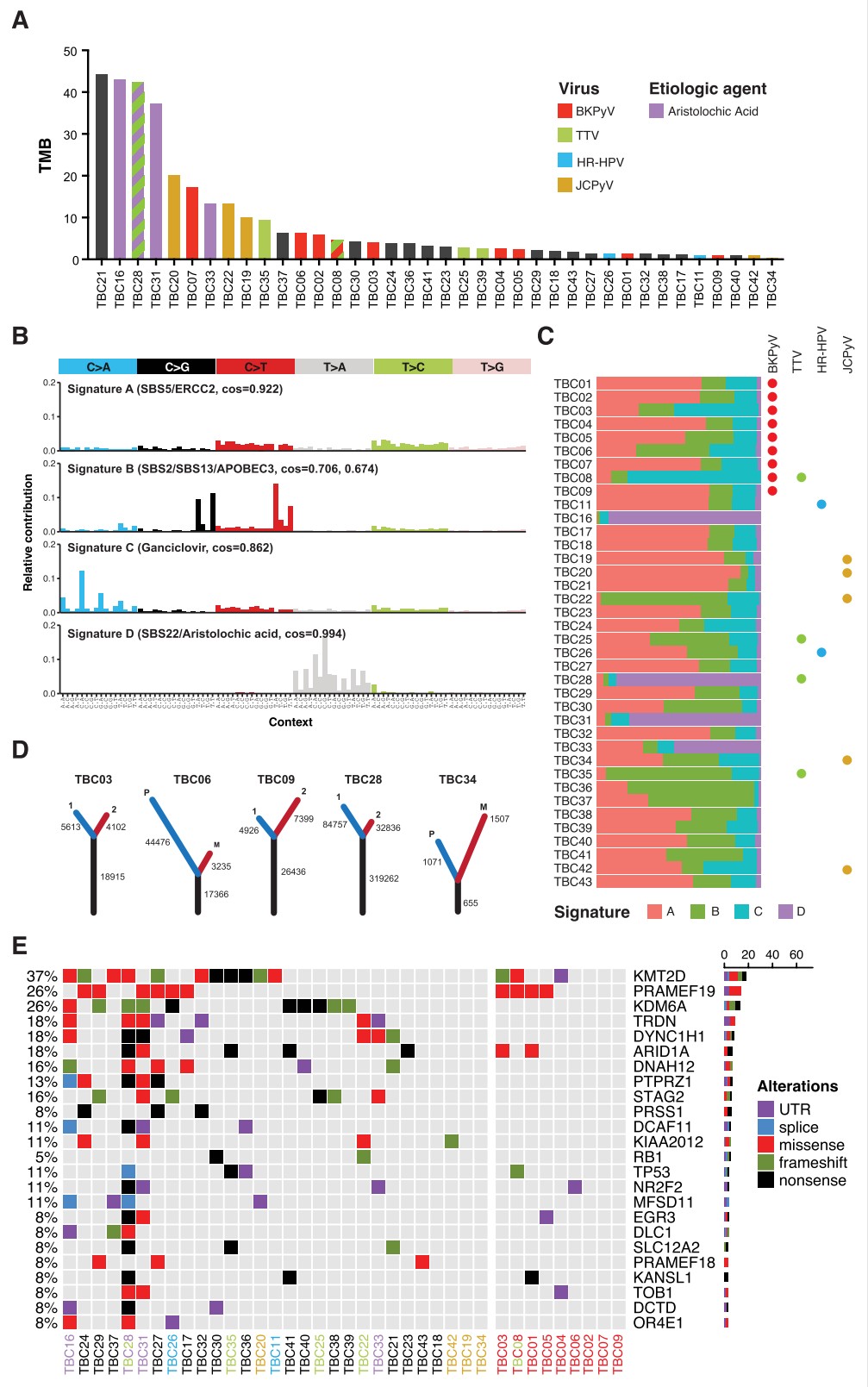

**Figure 6.** Somatic point mutations and mutation signature analysis. (**A**) Tumor mutation burden (TMB, non-synonymous mutations per million bases) for each tumor in this study. Bars are colored by viral positivity (red, BK polyomavirus (BKPyV); green, TTV; blue, HR-HPV; goldenrod, JC polyomavirus (JCPyV)) or etiologic agent (aristolochic acid, purple; black, undetermined). Multiple colors reflect multiple detected viruses or etiologies. (**B**)

*Figure 6 continued on next page*

*Figure 6 continued*

Barplots of the contribution of each trinucleotide substitution for the four deconvoluted signatures with the likely mutation process indicated. (**C**) Proportion of each deconvoluted signature that contributes to each sample with virus status indicated by colored circles (red, BKPyV; green, TTV; blue, HR-HPV; goldenrod, JCPyV). (**D**) Number of unique and common trunk mutations in primary-metastatic tumor pairs and tumors with multi-region sequencing. For TBC03, TBC09, and TBC28, branches one and two refer to two separate areas of the same tumor. For TBC06 and TBC34, branches P and M refer to the primary tumor and metastasis, respectively. (**E**) Oncoprint for the top mutated genes in bladder cancers of transplant patients. Tumors IDs are colored by likely etiology: BKPyV-positive, red; JCPyV-positive, goldenrod; HR-HPV-positive, blue; TTV-positive, green; aristolochic acid, purple; undetermined, black. The percent of modified tumors is shown on the left and the count of the variants in each gene is represented by the barplot on the right.

The online version of this article includes the following figure supplement(s) for figure 6:

**Figure supplement 1.** Mutations signature deconvolution.

and papillomaviruses can play a carcinogenic role in the development of bladder cancer, particularly among transplant recipients.

BKPyV infection in organotypic urothelial cell culture has been shown to promote cellular proliferation (*Schneidewind et al., 2020*). This is most likely through the transforming effects of viral T antigens, as is supported here and in previous studies by the loss of late region transcripts and enrichment of early region transcription in primary tumors, and maintenance of these transcripts in metastatic lesions (*Müller et al., 2018*). Interestingly, we observed frequent clonal loss of the p53-inactivating helicase domain of BKPyV LTag due to deletions and point mutations in the integrated virus. While such deletions in LTag are commonly observed in MCPyV-positive Merkel cell carcinoma (*Shuda et al., 2008*), MCPyV LTag lacks the p53-inactivating activity of the C-terminal helicase domain of BKPyV. One might thus have expected the C-terminal portion of BKPyV LTag to be preserved in tumor cells. We speculate that the loss of the BKPyV helicase domain is driven by negative selection against deleterious effects of LTag on tumor survival (e.g. LTag might unwind the integrated BKPyV origin of replication and initiate 'onion skin' DNA structures leading to chromosomal instability and cell death). The absence of the p53-binding domain may be compensated for in some BKPyV-positive tumors through the significantly increased expression of the ubiquitin ligase TRIM71 that we observed. TRIM71 has been shown to bind and poly-ubiquitinate p53 for proteasomal degradation and prevent apoptosis during stem cell differentiation (*Nguyen et al., 2017*).

We also observed amplification of the host genome surrounding BKPyV integration sites, consistent with circular DNA intermediates and/or MMEJ break-induced replication. Similar findings have been reported for HPV and MCPyV-associated tumors (*Starrett et al., 2020*; *Czech-Sioli et al., 2020*). These amplification events result in a variable number of tandem head-to-tail copies of the virus and host genome that are thought to create super-enhancers affecting viral and host gene expression (*Warburton et al., 2018*; *Dooley et al., 2016*). In cervical cancer, frequently only one integration event is transcriptionally active; however, in tumors carrying integrated BKPyV sequences, the abundance of viral DNA and RNA are positively correlated, suggesting that each integrated copy produces viral transcripts. While the observed integration sites in this study are unique and have not been observed in Merkel cell carcinoma, HPV16 integration has been reported previously in *BCAR3* (*Jeannot et al., 2018*). Elevated expression of *BCAR3* has been shown to increase the proliferation, motility, and invasiveness of estrogen receptor-positive breast cancer cells after treatment with anti-estrogens (*Wallez et al., 2014*; *Wilson et al., 2013*).

In five tumors harboring integrated BKPyV sequences (TBC02, TBC03, TBC05, TBC06, and TBC08), we observed significant upregulation of genes associated with cell cycle progression, DNA damage, histones, and the mitotic spindle. Tumors with evidence of BKPyV integration also exhibited significant downregulation of keratins and cell adhesion genes. The latter may contribute to the high-grade and invasive behavior of BKPyV-positive tumors observed in this study and others (*Alexiev et al., 2013*; *Sirohi et al., 2018*; *Kenan et al., 2015*; *Kenan et al., 2017*; *Nickeleit et al., 2018*).

Many of the observed gene expression changes are consistent with known effects of BKPyV infection and the specific activities of LTag, which binds Rb-family proteins and alters the active pool of E2F transcription factors in the cell (*Caller et al., 2019*; *Harris et al., 1996*). Recent studies have shown that APOBEC3B expression is repressed by the DREAM complex (which is composed of Rb-family proteins

and E2F transcription factors *Starrett et al., 2019*) and, accordingly, we found that APOBEC3B is more highly expressed in BKPyV-positive tumors compared to normal tissues and tumors without tumor virus sequences, likely due to LTag activity. However, despite this increased expression, the mutation signature commonly attributed to APOBEC3B did not appear enriched in BKPyV-positive tumors. It is possible that tumors expressing BKPyV LTag and increased APOBEC3B manifested greater intratumor mutational heterogeneity, but we were unable to detect possible low-frequency APOBEC3-mediated variants from FFPE tissue without deeper and more accurate sequencing. Additionally, consistent with the disruption of the DREAM complex in these tumors, we observed higher expression of *MYBL2*, a key component of the MMB complex, and one of its targets, *FOXM1*, which regulates numerous genes required for G2/M progression. We also observed increased expression of *FOXM1* downstream targets associated with the centromere and kinetochore, which have been shown to promote improper chromosome segregation and tumorigenesis (*Fischer and Müller, 2017*; *Sadasivam et al., 2012*; *Schade et al., 2019*).

BKPyV-positive tumors in our study had significantly higher expression of a number of genes that promote homologous recombination (e.g. *RAD51*, *RAD54L*, *BRCA1*, and *BRCA2*) and protect against replication fork stalling and collapse (e.g. *RAD51*, *XRCC2*, and *FANCB*) relative to virus-negative tumors (*Tye et al., 2021*). Claspin (*CLSPN*) and *TIMELESS*, which interact with replicative polymerases and helicases, are also highly expressed in BKPyV-positive tumors, further promoting replication fork progression and genome stability (*Bianco et al., 2019*). This expression pattern might promote cell survival in the face of genomic damage caused by viral genome integration, oncogene expression, and APOBEC3B upregulation.

While HPVs are not generally considered causative agents of bladder cancer, they have been detected in rare cases of bladder cancer affecting immunocompetent and immunosuppressed patients (*Zapatka et al., 2020*; *Cantalupo et al., 2018*; *Guma et al., 2016*). In the current study, we identified four tumors with carcinogenic *Alphapapillomavirus* sequences (HPV16 or HPV51). *Alphapapillomaviruses* are believed to cause cancer through the sustained expression of their E6 and E7 oncoproteins, which is frequently associated with the integration of the papillomavirus genome into the tumor genome.

One case in the panel carried sequences of HPV20, a *Betapapillomavirus* that can cause cutaneous squamous cell carcinoma in animal model systems (*Michel et al., 2006*). The possible involvement of *Betapapillomaviruses* in skin cancer in the general population remains controversial (*Viarisio et al., 2018*). In epidermodysplasia verruciformis, a rare syndrome caused by defects in zinc-binding proteins EVER1 and EVER2, patients frequently develop non-melanoma skin cancers containing *Betapapillomaviruses* (*Dell'Oste et al., 2009*). Expression of E6 and E7 from *Betapapillomaviruses* has been shown to promote cell survival in the face of ultraviolet radiation damage and other carcinogenic insults (*Michel et al., 2006*; *Viarisio et al., 2018*; *Viarisio et al., 2016*). In the context of bladder cancer, it is possible that cutaneous papillomaviruses likewise enable the accumulation of carcinogenic DNA damage. Additionally, the identification of HPV28, an *Alphapapillomavirus* that is not generally associated with cervical cancer, suggests more abundant papillomavirus infections of the bladder than previously assumed, with unknown implications for carcinogenesis.

An explanation for the observation that viruses are more prevalent in bladder cancers affecting solid organ transplant recipients compared to cases in the general population is that, in combination with immune suppression, transplant recipients may often become newly infected through transmission from the donor graft at the time of transplantation, perhaps with a different viral genotype than present in the host previously. This phenomenon is commonly observed in kidney transplantation and is associated with BKVN (*Solis et al., 2018*), but has not been documented for heart, lung, or liver transplant recipients, who are also included in the current study. Additionally, this study and one prior study (*Querido et al., 2020*) identified JCPyV in bladder tumors. Based on the high degree of similarity between JCPyV and BKPyV, it seems reasonable to expect that the two species would behave similarly. However, the low abundance of JCPyV RNA and DNA in these specimens and the absence of integration, together with the ubiquity of latent JCPyV infections in the urinary tract, raises the possibility that these observations reflect incidental detection events.

The data from this study and others suggest that in the context of strong immune the suppression BKPyV can cause bladder cancer through clonal integration but is rarely detected in tumors of the general population. While most adults are seropositive for BKPyV, with at least 10% having detectable

BKPyV in the urine, BKPyV is only observed in upwards of 4% of NMIBC cases and less than 0.25% in muscle-invasive bladder cancers in the general population (*Cantalupo et al., 2018*; *Llewellyn et al., 2018*). This implies that, while BKPyV LTag can provide a growth advantage to cells in culture, the large multi-domain antigen may be relatively immunogenic compared to the much smaller oncoproteins encoded by high-risk HPVs or the highly truncated MCPyV LTag isoforms typically observed in Merkel cell carcinomas. Immunologic recognition of these tumors may also be impacted by the increased expression of APOBEC3B, which can generate immunogenic neoantigens (*Serebrenik et al., 2019*; *Chen et al., 2019*). Several reports of regression of patients' BKPyV-positive tumors after reduction of immune suppression support the idea that tumors constitutively expressing BKPyV gene products are readily targeted and controlled by the immune system (*Meier et al., 2021*; *Cuenca et al., 2020*; *Fu et al., 2018*). The theoretical immunological costs of viral gene expression for a nascent tumor cell raise the possibility of 'hit-and-run' carcinogenesis. The hit-and-run hypothesis invokes the idea that a virus may play a causal role in the early stages of carcinogenesis but then become undetectable at more advanced stages of tumor development. Infection of a premalignant cell may promote its growth and survival through the expression of the viral oncogenes. Additionally, the expression of viral oncogenes may promote genome instability through the expression of the mutagenic APOBEC3 enzymes or other mechanisms that further push the cell towards transformation, as has been suggested by a recent study of BKPyV infection in differentiated urothelium (*Baker et al., 2022*).

The heterogeneous expression of LTag observed in this study could represent transcriptional silencing or loss of BKPyV DNA from one part of the tumor, supporting the idea that tumors can lose the need for LTag expression. Alternatively, our observations could be accounted for by a multi-stage integration and carcinogenesis process proposed by other recent studies on BKPyV-positive urinary tumors from kidney transplant recipients (*Jin et al., 2021*; *Wang et al., 2020*). However, our sequencing experiments support a dominant clonally integrated form likely established early during tumor development in most BKPyV-positive tumors in this study. The only exception to this observation is TBC01, which appears to exhibit a viable BKPyV episome with a rearranged regulatory region present in a small subset of tumor cells. This tumor likely represents a passenger infection of an existing tumor (*Dalianis and Hirsch, 2013*). Future studies should investigate the hypothesis that passenger infections might play an oncomodulatory role in tumor development. This also suggests that archetypal BKPyV, rather than the more pathogenic rearranged strains found in cases of nephropathy, is more likely to integrate and be preserved into nascent tumor cells. In support of the idea that integration may be a more common aspect of BKPyV infection than previously assumed, we identified a clonal BKPyV integrant in the normal bladder specimen from case TBC09 in both the RNA and WGS sequencing that was distinct from the BKPyV integrant observed in the tumor sample. The normal tissue integrant had multiple copies of small T antigen and a large deletion in the regulatory regions (*Figure 1—figure supplement 1*). Only a few reads from RNA sequencing mapped to the small T antigen region, and histology of the section indicates no tumor cells or LTag staining, suggesting that the virus did not integrate into the right genomic location or maintain the needed components to drive carcinogenesis.

It remains to be seen whether TTVs contribute to disease in the context of immune suppression. A general model is that these ubiquitous viruses establish a chronic infection that the immune system generally keeps in check, but immune suppression results in increases not only in the abundance but also in the diversity of TTVs observed in hosts (*De Vlaminck et al., 2013*). Indeed, the detection of TTVs can serve as an indicator of the degree of overall immune suppression in transplant recipients (*Blatter et al., 2018*). Interestingly, these viruses, like papillomaviruses and polyomaviruses, also appear to be depleted for APOBEC3 target motifs, consistent with the effects of an evolutionary virus-host arms race (*Poulain et al., 2020*; *Verhalen et al., 2016*; *Warren et al., 2015a*).

Until recently, this type of molecular assessment from FFPE tissues would have been nearly impossible or badly muddled by the highly damaging effects of formalin fixation and oxidation of nucleic acids over time. Recent advancements in the isolation of nucleic acids, such as low temperature and organic solvent-free deparaffinization, combined with efficient library preparation from low-concentration highly degraded sources, yielded sufficiently high-quality material for WGS variant calling and total RNA sequencing (*Robbe et al., 2018*). To address the difficulty of accurately calling somatic variants (which can be problematic even from flash-frozen or fresh tissues), we called variants using the consensus of three modern variant callers. The lack of matched normal tissues for

most cases is a limitation of this work, but our analytical approach accounted for this by focusing the analysis on mutations with >10% allele frequency and those with potential functional effects, and by excluding known germline variants. Our methods were validated internally through the sequencing of separate regions from the same tumor and of primary-metastatic pairs, which reveal similar concordance of mutations as has been reported from flash-frozen tissues (*Zhang et al., 2014*). Our variant calling approach was also validated by the observation that we detected four deconvoluted mutation signatures that match those expected from prior surveys of bladder cancer. However, the low overall coverage of our WGS remains a limitation of this study.

We identified four bladder cancers in kidney transplant recipients that exhibited abundant mutations attributable to aristolochic acid-mediated DNA adducts. Aristolochic acid is a highly nephrotoxic and mutagenic compound produced by birthwort plants, which sometimes contaminates certain types of herbal medicines and grains (*Poon et al., 2015*). Exposure to this compound likely contributed to the patients' need for kidney transplantation, as well as their eventual development of bladder cancer. Highlighting the highly mutagenic nature of this compound, the four cases with dominant aristolochic acid signatures were in the top seven for total mutation burden (*Figure 6*). None of the three tumors had detectable oncogenic viral sequences, but one had detectable TTV. We also identified likely ganciclovir-mediated mutations (*de Kanter et al., 2021*) in most patients indicating that this common treatment to prevent reactivation of cytomegalovirus in solid organ transplant recipients may promote mutagenesis in the urinary bladder. Unfortunately, ganciclovir treatment history was unavailable to confirm that this is the origin of this mutation signature in these cases. Ever-decreasing sequencing costs will facilitate additional studies of this type and shed light on rare and understudied tumor types, as well as analyses of lower-grade and pre-cancerous lesions.

## Materials and methods
### Sample acquisition and ethics
The Transplant Cancer Match (TCM) Study is a linkage of the US national solid organ transplant registry with multiple central cancer registries (https://transplantmatch.cancer.gov/). We used data from this linkage to identify cases of in situ or invasive bladder cancer diagnosed among transplant recipients. Staff at five participating cancer registries (California, Connecticut, Hawaii, Iowa, Kentucky) worked with hospitals in their catchment areas to retrieve archived pathology materials for selected cases.

We obtained twenty 10 micron sections from formal-fixed paraffin-embedded (FFPE) blocks for each specimen with available material. At each originating institution, the microtome blade was cleaned with nuclease-free water and ethanol between samples. Single 5 micron sections leading and trailing the twenty sections used for nucleic acid isolation were saved for histochemistry and one additional section was used for immunohistochemistry. Hematoxylin and eosin-stained sections were reviewed by a trained pathologist and tumor purity was determined by cellular morphology.

### Nucleic acid isolation
Samples were simultaneously deparaffinized and digested using 400 µL molecular-grade mineral oil (Millipore-Sigma) and 255 µL Buffer ATL (Qiagen) supplemented with 45 µL of proteinase K (Qiagen). Samples were incubated overnight at 65 °C in a shaking heat block. Samples were spun at 16,000 × g in a tabletop microcentrifuge for one minute to separate the organic and aqueous phases. Depending on the presence of visible remaining tissue, some samples were subjected to one or two additional 2 hr long digests by the addition of 25 µL of fresh proteinase K buffer. Lysates were stored at 4 °C until RNA or DNA isolation and processed within one month.

Lysates were spun at 16,000 × g in a tabletop microcentrifuge for one minute. For DNA isolation, 150 µL of supernatant was moved to a new 1.5 mL tube. 490 µL of binding buffer PM (Qiagen) and 10 µL of 3 M sodium acetate were added to the lysate. The mixture was then added to a Qiaquick spin column and spun at 16,000 × g for 30 s. Flow-through was reapplied to the spin column for complete binding. The column was washed first with 750 µL of Buffer PE (Qiagen) and then 750 µL of 80% ethanol, spinning at 16,000 × g for 30 s and discarding flow-through each time. The column was dried by spinning it at 16,000 × g for 5 min. Collection tubes were discarded, and the column was moved to a new microcentrifuge tube. 50 µL of pre-warmed, 65 °C 10% buffer EB was applied to the column and incubated for 1 min. The tube was then spun at 16,000 x g for 2 min. DNA quantity and quality were assessed by Qubit

(Thermo Fisher) and spectrophotometry (DeNovix). DNA was stored at –20 °C until used for library preparation. Only samples with greater than 50 ng of DNA were processed for library prep.

For RNA isolation, 150 µL of the remaining clarified lysate was moved to a new tube. 250 µL of buffer PKD (Qiagen) was added and vortexed to mix. The remainder of the RNA extraction process was carried out using an RNeasy FFPE Kit (Qiagen) according to the manufacturer's protocol. RNA quantity and quality were assessed by spectrophotometry (DeNovix) and TapeStation (Agilent).

## Immunohistochemistry

FFPE 5 µm thick tissue sections mounted on charged glass slides were stained with antibody against Large T Antigen, clone PAb416 (Sigma Millipore, cat. DP02), which detects LTag from multiple poly-omaviruses, including SV40, BKPyV, JCPyV, WU, KI, 6, 7, 10, and 11 (*Toptan et al., 2016*). Slides were baked in a laboratory oven at 60 °C for 1 hr prior to immunostaining on Ventana Discovery Ultra automated IHC stainer upon following conditions: CC2 (pH9) antigen retrieval for 64 min at 96 °C, antibody at concentration 0.5 µg/ml in Agilent antibody diluent (cat. S3022) for 32 min at 36 °C, Anti-Mouse HQ-Anti HQ HRP detection system for 12 min with DAB for 4 min and Hematoxylin II coun-terstain for 8 min. After washing per manufacturer's instructions, slides were incubated in tap water for 10 min, dehydrated in ethanol, cleared in xylene, coverslipped with Micromount media (Leica Biosystems), and scanned on AT2 slide scanner (Leica Biosystems) for pathology review. FFPE sections of cell pellets transfected with LTag and commercial slides of SV40 infected tissue (Sigma, cat. 351 S) were used as positive controls.

## Library preparation and sequencing

50–250 ng of isolated DNA was fragmented in microtube-50 using a Covaris sonicator with the following settings: peak power: 100, duty factor: 30, cycles/burst: 1000, time: 108 s. End-repair and A-tailing were performed on fragmented DNA using the KAPA HyperPrep Kit (Roche). NEB/Illumina adaptors were ligated onto fragments with KAPA T4 DNA Ligase for 2 hr at 20 °C then treated with 4 µL USER enzyme (NEB) for 15 min at 37 °C to digest uracil-containing fragments. Ligation reactions were cleaned up using 0.8 x AMPure XP beads using the KAPA protocol. NEB dual-index oligos were added to the adaptor-ligated fragments and amplified for 6–8 cycles (depending on the amount of input fragmented DNA) using KAPA HiFi HotStart ReadyMix (Roche). Final amplified libraries were cleaned using 1 x AMPure beads with the recommended KAPA protocol. Ribosomal sequence-depleted cDNA libraries were prepared using 50 ng of total RNA with the SMARTer Stranded Total RNA-Seq Kit v2 – Pico Input Mammalian (Takara) following the manufacturer's instructions for FFPE tissues. Final RNA and DNA libraries were assessed for size and quantity by Agilent TapeStation. Only samples that yielded libraries greater than 5 nM were sequenced.

DNA libraries were sequenced on the Illumina NovaSeq 6000 at the Center for Cancer Research (CCR) Sequencing Facility. RNA libraries were sequenced on the Illumina NovaSeq 6000 and NextSeq 550 in high output mode at the CCR Genomics Core. Sequencing metrics are reported in *Supplementary file 1b*.

## Sequence alignments

Reads were trimmed using Trim Galore 0.6.0 with default settings. RNA reads was initially aligned using STAR aligner 2.5.3ab (*Dobin et al., 2013*) against a fusion reference human genome containing hg38, all human viruses represented in RefSeq as of December 2018 (*Supplementary file 1c*), and all papillomavirus genomes from PaVE https://pave.niaid.nih.gov (*Van Doorslaer et al., 2017*). Default parameters were used with the following exceptions: chimSegmentMin = 50, outFilterMultimapNmax = 1200, outFilterMismatchNmax = 30, outFilterMismatchNoverLmax = 1. Any reads that had less than 30 bp of perfect identity were excluded. Trimmed DNA reads were aligned with Bowtie2 (2.3.4.3) using the `--very-sensitive` setting to the same reference genome as mentioned above excluding RNA viruses (*Langmead and Salzberg, 2012*). Alignments were sorted and duplicate sequences were flagged using Picard 2.20.5. Indel realignments and base quality recalculations were conducted using GATK.

## Virus detection and integration analysis

All WGS reads not mapping to the human genome were de novo assembled using MEGAHIT (1.1.4) with default parameters (*Li et al., 2015*). All trimmed RNA reads were assembled using

RNASPAdes (*Bushmanova et al., 2019*). Assembled contigs were annotated using BLASTn and BLASTx against the NCBI nt database (October 2021) for closely related species, and Cenote-Taker2 version 2.1.2 (https://github.com/mtisza1/Cenote-Taker2; *Tisza, 2021*) was used to identify more divergent species in contigs ≥1000 bp (*Tisza et al., 2021*). Depth and breadth of coverage of viral species were normalized by the total number of human reads and length of the viral genome. Only species with ≥10% genome coverage and a normalized depth ≥10 for a viral genome in a given sample were considered as hits. Viral read k-mers were cross-compared against samples for uniqueness to identify index hopping or potential contamination between samples. Rearrangements in the BKPyV regulatory region were analyzed and annotated using BKTyper (*Martí-Carreras et al., 2020*).

Bam alignments were input into Oncovirus Tools (https://github.com/gstarrett/oncovirus_tools) to call integration sites (*Starrett et al., 2020*; *Starrett, 2020*). It starts by extracting discordant read pairs (where one read aligns to a sequence of interest, i.e., virus, and the mated read aligns to the human genome) and any remaining reads aligned to the human genome that contain at least one 25 bp k-mer from the input sequences of interest as determined by a Bloom filter. It uses the human genomic coordinates from the above reads to identify putative integration regions by merging their stranded mapping positions to find overlaps, counting the number of reads per stranded region. Oncovirus Tools then assembles the extracted reads, together with all unaligned reads, using Spades (*Bankevich et al., 2012*). The resulting assembly graphs are annotated with the human and viral genomes using BLASTn and the annotated assembly graphs are plotted using the R package ggraph. The output is then screened for contigs containing both human and viral hits with BLASTn e-values below 1e-10. Based on these hits, integration junctions are called and overlaps in host-virus hit on the contigs are then screened for microhomology. All putative integration sites from Oncovirus Tools were manually validated by returning to the original alignment file.

## Transcriptome clustering and differential gene expression analysis

Counts from STAR were input into R and normalized using the DESeq2 vst function (*Love et al., 2014*). The DESeq2 model was built using the following factor: tissue type (normal, primary, metastasis), grade, stage, and virus status to evaluate their effects on gene expression. Since the RNA seq libraries were prepared in different batches on different days and in different sequencing runs, batch effects were removed using the R package limma and the function RemoveBatchEffects. These normalized counts were input into the R package ConsensusClusterPlus. Pathway analysis was conducted using Enrichr (https://amp.pharm.mssm.edu/Enrichr) (*Kuleshov et al., 2016*; *Chen et al., 2013*).

## Somatic point mutation, structural variant, and copy number variant calling

Point mutations were called using Mutect2, VarScan2, and lofreq with default parameters (*Koboldt et al., 2012*; *Wilm et al., 2012*). Consensus calls between these variant callers were performed using SomaticSeq (3.3.0) (*Fang et al., 2015*). Likely germline variants were annotated and removed using SnpSift and dbSNP v152. Likely somatic point mutations were further filtered by the following criteria: SomatiqSeq PASS filter, ≥10% allele frequency, ≥4 reads supporting the variant allele, and ≥8 reads of total coverage of that position. Common mutations in cancer were annotated using SnpSift and COSMIC. Somatic mutations enrichment by gene was determined using the R package dNdScv. Copy number variants in tumor WGS datasets were called using GATK4 CNV to compare them to a panel comprised of the normal-tissue WGS datasets generated in this study. Recurrent copy number variants within polyomavirus-containing tumors or tumors with no virus were determined using GISTIC2 with default parameters. Visualization and variant calling for BKPyV were performed on alignments against a BKPyV genotype Ib-2 isolate (accession number: AB369087.1).

## Mutation signature analysis

Mutation signature analysis was conducted using the likely somatic variants passing all the above criteria. Mutational Patterns and Somatic Signatures R packages were used for de novo somatic mutations signature analysis.

## Data visualization

All graphs were made in the R statistical environment (4.0.3) using the package ggplot2 or using GraphPad Prism.

# Acknowledgements

We would like to thank the CCR Genomics Core and CCR Sequencing Facility for their assistance with sequencing. This work utilized the computational resources of the NIH HPC Biowulf cluster (http://hpc.nih.gov). We would also like to thank the members of the Laboratory of Cellular Oncology for their useful discussion and insights. We would like to the staff at the Scientific Registry of Transplant Recipients and participating cancer registries who assisted with the collection of the data and registry linkages.

# Additional information

### Competing interests

Yelena Golubeva, Petra Lenz: is affiliated with Leidos Biomedical Research Inc. The author has no financial interests to declare. Reuben S Harris: is a co-founder, shareholder, and consultant of ApoGen Biotechnologies Inc. The other authors declare that no competing interests exist.

### Funding

| Funder | Grant reference number | Author |
| --- | --- | --- |
| National Institutes of Health | Intramural Research Program | Gabriel J Starrett<br>Eric A Engels<br>Christopher B Buck |

The funders had no role in study design, data collection and interpretation, or the decision to submit the work for publication.

### Author contributions

Gabriel J Starrett, Conceptualization, Data curation, Formal analysis, Validation, Investigation, Visualization, Methodology, Writing – original draft, Writing – review and editing; Kelly Yu, Data curation, Validation, Project administration, Writing – review and editing; Yelena Golubeva, Mary L Piaskowski, Investigation, Writing – review and editing; Petra Lenz, Formal analysis, Validation, Writing – review and editing; David Petersen, Validation, Investigation, Writing – review and editing; Michael Dean, Ajay Israni, Brenda Y Hernandez, Thomas C Tucker, Iona Cheng, Lou Gonsalves, Cyllene R Morris, Shehnaz K Hussain, Charles F Lynch, Reuben S Harris, Ludmila Prokunina-Olsson, Paul S Meltzer, Resources, Writing – review and editing; Christopher B Buck, Conceptualization, Resources, Supervision, Funding acquisition, Writing – original draft, Project administration, Writing – review and editing; Eric A Engels, Conceptualization, Resources, Formal analysis, Supervision, Funding acquisition, Writing – original draft, Project administration, Writing – review and editing

### Author ORCIDs

Gabriel J Starrett http://orcid.org/0000-0001-5871-5306
Mary L Piaskowski http://orcid.org/0000-0001-8453-6416
Reuben S Harris http://orcid.org/0000-0002-9034-9112
Ludmila Prokunina-Olsson http://orcid.org/0000-0002-9622-2091
Christopher B Buck http://orcid.org/0000-0003-3165-8094

### Ethics

The TCM Study is considered non-human subjects research at the National Institutes of Health because researchers do not receive identifying information on patients, and the present project utilizes materials collected previously for clinical purposes. The TCM Study was reviewed, as required, by human subjects committees at participating cancer registries.

Decision letter and Author response
Decision letter https://doi.org/10.7554/eLife.82690.sa1
Author response https://doi.org/10.7554/eLife.82690.sa2

## Additional files

### Supplementary files

• Supplementary file 1. Supporting information. (a) Checkerboard table of samples used in this study. (b) Sequencing metrics. (c) Reference sequences used in this study. (d) Tumor torque teno virus similarities. (e) BKPyV-positive tumor vs virus-free tumor significantly differentially expressed genes. (f) Non-synonymous point mutations. (g) Copy number variants.

• MDAR checklist

### Data availability

All refseqs for human papillomaviruses were downloaded from PaVE and refseqs for human polyomaviruses were downloaded from NCBI as of November 2018. All sequencing data generated in this study are available from dbGaP under accession phs003012.v1.p1. Viral contigs from this study are deposited in GenBank under accessions OQ469311 and OQ469312. All other contigs and larger IHC images are deposited in figshare (https://figshare.com/projects/Common_Mechanisms_of_Virus-Mediated_Oncogenesis_in_Bladder_Cancers_Arising_in_Solid_Organ_Transplant_Recipients/132833). Code used in this manuscript are available from https://github.com/gstarrett/oncovirus_tools (*Starrett, 2020*).

The following datasets were generated:

| Author(s) | Year | Dataset title | Dataset URL | Database and Identifier |
|---|---|---|---|---|
| Starrett GJ | 2022 | Multi-omics of Bladder Cancers of Solid Organ Transplant Recipients | https://www.ncbi.nlm.nih.gov/projects/gap/cgi-bin/study.cgi?study_id=phs003012.v1.p1 | NCBI dbGaP, phs003012.v1.p1 |
| Starrett GJ | 2023 | pAb416 IHC images | https://doi.org/10.6084/m9.figshare.19199723.v1 | figshare, 10.6084/m9.figshare.19199723.v1 |
| Starrett GJ | 2023 | Anelloviridae contigs | https://doi.org/10.6084/m9.figshare.22124240.v1 | figshare, 10.6084/m9.figshare.22124240.v1 |
| Starrett GJ | 2023 | Anellovirus sp. isolate TBC35T_k141_20313 ORF2/4 (ORF2/4) gene, complete cds; and ORF1 (ORF1) gene, partial cds | https://www.ncbi.nlm.nih.gov/nuccore/OQ469311 | NCBI Nucleotide, OQ469311 |
| Starrett GJ | 2023 | Anellovirus sp. isolate TBC35T_k141_22964 ORF1 (ORF1) gene, partial cds; and ORF2 (ORF2) gene, complete cds | https://www.ncbi.nlm.nih.gov/nuccore/OQ469312 | NCBI Nucleotide, OQ469312 |

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
