## [Editor Report]

This work provides a compelling case that viral origins of bladder cancer should be more carefully considered. Specifically, the clonogenic expression of viral oncogenes in these tumors combined with the lower than expected prevalence of major tumor suppressor (p53 and pRB) provides strong evidence for the authors' assertions. Ultimately, it will be important to follow up this work and I look forward to seeing those next steps.

---

## [Decision Letter]

**Decision letter after peer review:**

Thank you for submitting your article "Evidence for Virus-Mediated Oncogenesis in Bladder Cancers Arising in Solid Organ Transplant Recipients" for consideration by *eLife*. Your article has been reviewed by 2 peer reviewers, including Nicholas Wallace as Reviewing Editor and Reviewer #2, and the evaluation has been overseen by Wafik El-Deiry as the Senior Editor.

Essential revisions:

(1) Despite showing strong evidence that viruses are altering the tumor cell environment, it is unclear if these changes are necessary for tumorigenesis. The heterogeneity of the LT expression suggests that the presence of the viral DNA and RNA may not be enough to assess whether it is actively contributing to the tumor. Is an increased frequency of viral protein staining linked with any evidence of an active contribution to tumorigenesis (fewer tumor-suppressor/oncogene mutations). that they reduced mutations in tumor suppressors. This might be easiest to assess with the tumors that have oncogenic HPV DNA. (If those tumors lacked p53 and RB mutations, it would support a causative role for the virus.) For BKPyV+ cancers, is the virus playing a role as a driver, hit and run, or by-stander as described in Dalianis et al., 2013 (https://pubmed.ncbi.nlm.nih.gov/23357733/).

(2) The data presented here and previously published data (https://doi.org/10.1002/path.5012) hint at Chromosome 1 as a potential "hot spot" for viral integration. Alternatively, this could be due to the larger relative size of the chromosome or some other feature of the chromosome. This should be discussed.

(3) There are multiple places where the organization and presentation of the data needs to be improved. These are noted in individual reviewer comments provided below.

*Reviewer #1 (Recommendations for the authors):*

In general, there seem to be multiple mistakes in terms of sample numbers, figure references and figure legends, which complicated the review extensively. In addition, in many of the shown data, it was not clear why only specific samples have been chosen or why some samples have just been left out or have been "forgotten". Moreover, for some of the statements in the Results sections, literature references are missing. Abbreviations should always be explained and be consistent throughout the manuscript and according to current state of the art (DeCaprio, J. A., et al. (2020). Fields Virolgy: DNA Viruses). All of these points need to be properly addressed before publishing the manuscript.

Title:

The title should be adapted as data does not support "viral-mediated oncogenesis".

Related to Table 1:

The table should be adapted for a better overview with characteristics and statistics being clearly labeled (see below). Suggest structuring it as follows: age in years at diagnosis, years from transplant to diagnosis, race, sex, transplanted organ, grade, summary stage (also following text).

Characteristic Statistic

Median IQR

Age in years at diagnosis

N %

Race

In the category "race" seems 1 patient missing as the sum only results in 42 cases. Please adapt the table and text.

Page 7 line 3-6:

It is not clear why not 43 primary tumors have been analyzed (corresponding to Table 1 and Figure 1)? Why are numbers between WGS and total RNAseq different? From Figure 1 it seems that multiple samples have been used from one patient, is this correct? If yes, this is not clear in the results description, please adapt. What does normal tissue mean = healthy?

Page 7 line 18 and Figure S1:

Please reference Figure S1 in the text. Please show coverage plots of TBC07 and separate runs of TBC03 (as for TBC09).

Figure 1 and Materials and methods:

What does "no data" mean? Was there no material available or was the quantity/quality of DNA not enough? Please clarify this in the text and Materials and methods.

"We obtained twenty 10-micron sections from formal-fixed paraffin-embedded (FFPE) blocks for cases with available material." How does that fit with Table 1 talking about 43 primary tumors?

How do the authors explain that for some tumors DNA signatures but no latency dependent transcripts could be detected? Please integrate in the text.

Material and methods:

How long were the DNA and RNA samples stored at 4{degree sign}C? Prolonged storage at 4{degree sign}C, especially for RNA samples, could contribute to low sample quality. Please adapt in the material and methods.

Which polyomavirus LTags are recognized by the used antibody? Please mention.

Related to Table 2:

Please sort Table 2 according to the most frequent chromosome location so overrepresentation of e.g., chromosome 1 is obvious. Might this integration sites be a contamination or a real hot-spot? Please discuss.

All abbreviations used in the table should be explained.

Page 9 line 10:

How was tumor purity immunohistochemically assessed? Should be explained in the Materials and methods section.

Figure 4B:

How were the stabilized counts calculated? This should be described in the Materials and methods.

Was TBC01 included in the BKPyV positive tumors? There should be only 9 BKPyV-positive tumors rather than 10. The authors should show one graph with exclusion of TBC01.

Figure S2:

Wrong figure reference and figure legend. It is very hard to interpret the data if figure references and legends are not correctly cited. Authors should carefully review the whole manuscript.

Page 11 lines 9-15:

Is there any figure associated to the statement? It is unclear what this result means.

Figure 5:

TBC02 seems to be missing in the BKPyV positive tumors. All BKPyV-negative tumors and all 43 primary tumors should be displayed.

Page 12 lines 6-12 and Figure S3:

Paragraph should refer to Figure S3 and not S4. "A possible HPV51 integration site…": there is no data showing that.

Figure 6A:

What are the black bars representing? Why is TBC36 now BKPyV positive? Why is Figure 1 showing EBV but Figure 6 not? Why is Figure 6 showing TTV but Figure 1 not? Please explain and adapt.

Figure 6B:

Please show this analysis also for HPV-positive tumors.

Figure 6E:

It is suggested that the results are sorted according to the percentages and to cluster all BKPyV-positive tumors together.

Page 16 lines 1-22:

Paragraph about APOBUCK3B seems to be overrepresented in the discussion and consistency in statements failed. Some information is non-informative or even highly speculative. This paragraph should be extensively revised.

---

## [Author Response]

Essential revisions:(1) Despite showing strong evidence that viruses are altering the tumor cell environment, it is unclear if these changes are necessary for tumorigenesis. The heterogeneity of the LT expression suggests that the presence of the viral DNA and RNA may not be enough to assess whether it is actively contributing to the tumor. Is an increased frequency of viral protein staining linked with any evidence of an active contribution to tumorigenesis (fewer tumor-suppressor/oncogene mutations). that they reduced mutations in tumor suppressors. This might be easiest to assess with the tumors that have oncogenic HPV DNA. (If those tumors lacked p53 and RB mutations, it would support a causative role for the virus.) For BKPyV+ cancers, is the virus playing a role as a driver, hit and run, or by-stander as described in Dalianis et al., 2013 (https://pubmed.ncbi.nlm.nih.gov/23357733/).

We thank the reviewers for their thoughtful review. Viral oncoprotein heterogeneity has previously been reported for HPV and MCPyV and can be affected by both methodology and biology, for example tumor hypoxia has been shown to affect E6 and E7 expression. Our evidence for clonal integration in primary tumors and a metastasis and recurrent helicase-inactivating mutations support selection in tumorigenesis. As suggested, in Figure 6 we indeed show that no BKPyV-positive or HPV-positive tumor harbored mutations in RB1. Additionally, only one BKPyV-positive tumor and none of the HPV-positive tumors had a mutation in TP53. We have added further emphasis to this point on page 14,

“None of the HPV-positive tumors with WGS harbored mutations in *TP53* or *RB1*. Similarly, none of the polyomavirus-positive tumors harbored mutations in *RB1* and only TBC08 had a frameshift mutation in *TP53*.”

We have also added speculation about the unique aspects of TBC01:

“This tumor likely represents a passenger infection of an existing tumor. Future studies should investigate the hypothesis that passenger infections might play an oncomodulatory role in tumor development.”

(2) The data presented here and previously published data (https://doi.org/10.1002/path.5012) hint at Chromosome 1 as a potential "hot spot" for viral integration. Alternatively, this could be due to the larger relative size of the chromosome or some other feature of the chromosome. This should be discussed.

In the current study, the events on chromosome 1 do not represent a hotspot since all of the integration events are from one patient and within 1000 bp of each other. These events instead appear to represent a single integration event with complex rearrangements. Two observations of integration on this large chromosome is not suggestive of a hotspot.

(3) There are multiple places where the organization and presentation of the data needs to be improved. These are noted in individual reviewer comments provided below.

We have addressed this and corrected the supplemental table order, which was due to a previous version being uploaded with the submission. Please see the detailed responses to individual comments.

Reviewer #1 (Recommendations for the authors):In general, there seem to be multiple mistakes in terms of sample numbers, figure references and figure legends, which complicated the review extensively. In addition, in many of the shown data, it was not clear why only specific samples have been chosen or why some samples have just been left out or have been "forgotten". Moreover, for some of the statements in the Results sections, literature references are missing. Abbreviations should always be explained and be consistent throughout the manuscript and according to current state of the art (DeCaprio, J. A., et al. (2020). Fields Virolgy: DNA Viruses). All of these points need to be properly addressed before publishing the manuscript.

We apologize for this error and thank Reviewer #1 for pointing it out. It appears that a previous version of the supplemental material was uploaded with this manuscript, which led to most supplemental materials being off by one. We have checked the text regarding the issues raised by the reviewer, and we added references and defined abbreviations where we thought these changes were indicated.

Title:The title should be adapted as data does not support "viral-mediated oncogenesis".

We agree with Reviewer #1’s view that the data can be seen as falling short of conclusive proof of virus-mediated oncogenesis. But the finding of clonally integrated viruses in tumors and their metastases, the detection of distinctive transcriptomic signatures, and the absence of mutations in TP53 and RB1 – are accepted traditional lines of evidence for virus-mediated oncogenesis (for more detail, see: https://pubmed.ncbi.nlm.nih.gov/31336246/). We believe that starting the title with “Evidence for…” provides readers with an accurate description of the type of data that the manuscript is reporting without indicating that we are providing definitive proof.

Related to Table 1:The table should be adapted for a better overview with characteristics and statistics being clearly labeled (see below). Suggest structuring it as follows: age in years at diagnosis, years from transplant to diagnosis, race, sex, transplanted organ, grade, summary stage (also following text).Characteristic StatisticMedian IQRAge in years at diagnosisN %Race

We have reordered the table to bundle the common patient statistics (i.e. N % and Median IQR, respectively) together and clearly labeled the columns above the statistics.

In the category "race" seems 1 patient missing as the sum only results in 42 cases. Please adapt the table and text.

This has been corrected.

Page 7 line 3-6:It is not clear why not 43 primary tumors have been analyzed (corresponding to Table 1 and Figure 1)? Why are numbers between WGS and total RNAseq different? From Figure 1 it seems that multiple samples have been used from one patient, is this correct? If yes, this is not clear in the results description, please adapt. What does normal tissue mean = healthy?

As described in the Materials and methods on page 22, lines 24-25 and page 24, lines 9-10 samples that didn’t yield enough nucleic acids for library preparation were not sequenced. This should hopefully be clearer with the addition of the suggested Supplemental table 1 (supplementary file 1a) checkerboard. Please see Results page 7, line 18-19, and Figure 2 legend. We have added, on page 7 line 4, an additional definition of normal as “histologically non-malignant.” The legends for Figures 2 and 6 also explain that for samples TBC03, TBC09, and TBC28 two separate sections of the tumor were sequenced.

Page 7 line 18 and Figure S1:Please reference Figure S1 in the text. Please show coverage plots of TBC07 and separate runs of TBC03 (as for TBC09).

This has been added.

Figure 1 and Materials and methods:What does "no data" mean? Was there no material available or was the quantity/quality of DNA not enough? Please clarify this in the text and Materials and methods.

See the response to comment 4 and changes on Page 7 lines 3-6.

"We obtained twenty 10-micron sections from formal-fixed paraffin-embedded (FFPE) blocks for cases with available material." How does that fit with Table 1 talking about 43 primary tumors?

This text is describing the amount of sample collected per case. This has been revised to read:

“We obtained twenty 10-micron sections from formal-fixed paraffin-embedded (FFPE) blocks for each specimen with available material.”

How do the authors explain that for some tumors DNA signatures but no latency dependent transcripts could be detected? Please integrate in the text.

We are uncertain about what the reviewer means by “DNA signatures” and “latency-dependent transcripts.” If we infer that some tumors have mutational signatures associated with viruses but do not have viral transcripts, our sense – as stated in the Discussion – is that this might represent transient suppression of viral transcription or it could reflect hypothetical hit-and-run effects.

Material and methods:How long were the DNA and RNA samples stored at 4{degree sign}C? Prolonged storage at 4{degree sign}C, especially for RNA samples, could contribute to low sample quality. Please adapt in the material and methods.Which polyomavirus LTags are recognized by the used antibody? Please mention.

The manuscript has been revised to note that all samples were processed within one month of receipt into our laboratory. Of note, with FFPE source material and this extraction method, RNAs are already highly fragmented immediately after extraction. We have observed that fragmentation does not significantly increase with storage time, as RNA integrity numbers are consistently between 2 to 4 immediately after RNA extraction. Additionally, the Takara total RNAseq kit used here takes advantage of the fragmented nature of the input RNA by not including a fragmentation step for FFPE RN. Subsequent library size selection removes very small fragments.

For the PAb416 antibody used for IHC, we have added the following text to the methods:

“clone Pab416 (Σ Millipore, cat. DP02), which detects LTag from multiple polyomaviruses, including SV40, BKPyV, JCPyV, WU, KI, 6, 7, 10, and 11 (84).”

Related to Table 2:Please sort Table 2 according to the most frequent chromosome location so overrepresentation of e.g., chromosome 1 is obvious. Might this integration sites be a contamination or a real hot-spot? Please discuss.All abbreviations used in the table should be explained.

We appreciate the comment, but we feel that sorting by chromosome will make the table difficult to interpret. Currently, it is sorted by patient ID, which we hope will make it easier for readers to recognize that all of the chromosome 1 integration junctions are from the same patient and within about 1000 basepairs of each other. We do not infer that chromosome 1 is a common hotspot; rather, within that patient there is a complex rearrangement that occurred on chromosome 1. Abbreviations have been added.

Page 9 line 10:How was tumor purity immunohistochemically assessed? Should be explained in the Materials and methods section.

Tumor purity was determined by H and E staining and review by our pathologist. A description has been added to Materials and methods, page 22 line 26 and page 10, line 14.

Figure 4B:How were the stabilized counts calculated? This should be described in the Materials and methods.

Our method for stabilizing the counts is described in Figure 4 legend: “Variance stabilized counts for APOBEC3B from DESeq2.” and in Material and Methods page 26, line 20.

Was TBC01 included in the BKPyV positive tumors? There should be only 9 BKPyV-positive tumors rather than 10. The authors should show one graph with exclusion of TBC01.

This plot also contained the metastatic tumor of TBC06. We have now removed the TBC06 metastatic tumor and highlighted TBC01 in this plot so readers can discern its difference versus the other BKPyV-positive tumors.

Figure S2:Wrong figure reference and figure legend. It is very hard to interpret the data if figure references and legends are not correctly cited. Authors should carefully review the whole manuscript.

We thank Reviewer #1 for catching our inadvertent uploading of the wrong Supplementary files during the submission process. The correct supplements have been uploaded for the revision.

Page 11 lines 9-15:Is there any figure associated to the statement? It is unclear what this result means.

We’ve added Figure 4-supplement 2 to support this statement. Virus integration commonly increases the expression of adjacent host genes, which can be an important selective pressure for a particular integration event. However, our one case with observed increased RNA coverage seems to originate from an amplification of a relatively small section host DNA rather than increased expression of the entire gene.

Figure 5:TBC02 seems to be missing in the BKPyV positive tumors. All BKPyV-negative tumors and all 43 primary tumors should be displayed.

TBC02 has been added to panel B. All additional primary tumors with WGS have been added as panel E.

Page 12 lines 6-12 and Figure S3:Paragraph should refer to Figure S3 and not S4. "A possible HPV51 integration site…": there is no data showing that.

We apologize for the misordered supplement file. We have now added a panel showing evidence for HPV51 integration to Figure S5.

Figure 6A:What are the black bars representing? Why is TBC36 now BKPyV positive? Why is Figure 1 showing EBV but Figure 6 not? Why is Figure 6 showing TTV but Figure 1 not? Please explain and adapt.

Black bars indicate that there is no determined etiologic agent or virus, which has now been added to the figure legend. Some of the bars were incorrectly colored in 6A and have been corrected. The calls and colors in 6C and 6E are correct. TTV was detected through a different method to look for divergent sequences, so it cannot readily be shown in Figure 1; this is explained in the results and methods. The very low depth and percent coverage of the EBV reads detected in TBC07 and TBC03 tumors and absence of RNA reads suggest that these are likely incidental infections; therefore, we did not classify these tumors as EBV positive in Figure 6.

Figure 6B:Please show this analysis also for HPV-positive tumors.

This analysis was conducted on all tumors for which we were able to generate WGS data, including the HPV-positive tumors.

Figure 6E:It is suggested that the results are sorted according to the percentages and to cluster all BKPyV-positive tumors together.

The results are sorted by the calculated selection enrichment by dNdScv R package (more details have been added to the methods on page 27), which takes into account total synonymous, missense, indel, and nonsynonymous mutations in a sample and gene sizes. This is an important calculation because simply analyzing mutations by frequency almost always results in the identification of very large genes (TTN is a key example) rather than finding loci with evidence for positive selection.

Page 16 lines 1-22:Paragraph about APOBUCK3B seems to be overrepresented in the discussion and consistency in statements failed. Some information is non-informative or even highly speculative. This paragraph should be extensively revised.

In our view, an important purpose of scientific Discussion sections is to present plausible ideas that might help readers generate experimentally testable hypotheses and start building on the results. APOBEC3B has been shown to be upregulated by BKPyV infection and specifically by the activity of the suppression of the DREAM complex by LTag. APOBEC3-associated mutations are the second most abundant mutagenic process in all cancers and also the dominant mutation signature in bladder cancers arising in the general population. Furthermore, APOBEC3 mutagenesis has shaped the evolution of the BKPyV genome. It is of considerable interest to the field whether BKPyV infection contributes to APOBEC3 mutagenesis of the host genome in infected cells (see PMID 35194151). We see this as an important future direction for the field. For these reasons, we believe it is appropriate to provide a thorough discussion on APOBEC3B and have therefore not shortened the text.